# Rendering Wireless Environments Useful for Gradient Estimators: A Zero-Order Stochastic Federated Learning Method

## Abstract

Federated learning (FL) is a novel approach to machine learning that allows multiple edge devices to collaboratively train a model without disclosing their raw data. However, several challenges hinder the practical implementation of this approach, especially when devices and the server communicate over wireless channels, as it suffers from communication and computation bottlenecks in this case. By utilizing a communication-efficient framework, we propose a novel zero-order (ZO) method with two types of gradient estimators, one-point and two-point, that harnesses the nature of the wireless communication channel without requiring the knowledge of the channel state coefficient. It is the first method that includes the wireless channel in the learning algorithm itself instead of wasting resources to analyze it and remove its impact. The two main difficulties of this work are that in FL, the objective function is usually not convex, which makes the extension of FL to ZO methods challenging, and that including the impact of wireless channels requires extra attention. However, we overcome these difficulties and comprehensively analyze the proposed zero-order federated learning (ZOFL) framework. We establish its convergence theoretically, and we prove a convergence rate of $O(\frac{1}{\sqrt[3]{K}})$ with the one-point estimate and $O(\frac{1}{\sqrt{K}})$ with the two-point one in the nonconvex setting. We further demonstrate the potential of our algorithms with experimental results, taking into account independent and identically distributed (IID) and non-IID device data distributions.

## 1 Introduction

Zero-order (ZO) methods are a subfield of optimization that assume that first-order (FO) information or access to function gradients is unavailable. ZO optimization is based on estimating the gradient using function values queried at a certain number of points. The number of function queries depends on the assumptions of the problem. For example, in multi-point gradient estimates (Duchi et al., 2015; Agarwal et al., 2010), they construct the gradient by performing the difference of function values obtained at many random or predefined points. However, they assume that the stochastic setting stays the same during all these queries. For example, for functions $\theta \mapsto f(\theta, S)$ subject to a stochastic variable $S$, two-point gradient estimates have the form,

$$g = d\frac{f(\theta + \gamma\Phi, S) - f(\theta - \gamma\Phi, S)}{2\gamma}\Phi,$$

with $\theta \in \mathbb{R}^d$ the optimization variable, $\gamma > 0$ a small value, and $\Phi$ a random vector with a symmetric distribution. By contrast, one-point estimates that use only one function value (Flaxman et al., 2004; Li & Assaad, 2021; Mhanna & Assaad, 2022), principally obtained at a random point,

$$g = \frac{d}{\gamma}f(\theta + \gamma\Phi, S)\Phi,$$

assume that the settings are continuously changing during optimization. This is an important property as it resonates with many realistic applications, like when the optimization is performed in wireless environments or is based on previous simulation results. Recently, an appeal to ZO optimization

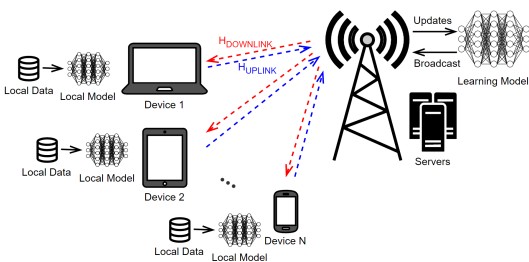

Figure 1: Federated learning over wireless networks.

is emerging in the machine-learning community, where optimizers are based on gradient methods. Examples include reinforcement learning (Vemula et al., 2019; Malik et al., 2019), generating contrastive explanations for black-box classification models (Dhurandhar et al., 2019), and effecting adversarial perturbations on such models (Ilyas et al., 2018; Chen et al., 2019).

On the other hand, with the massive amounts of data generated or accessed by mobile devices, a growing research interest in both sectors of academia and industry (Bonawitz et al., 2019) is focused on federated learning (FL) (McMahan et al., 2017), as it's a practical solution for training models on such data without the need to log them to a server. A lot of effort has been invested in developing first-order (McMahan et al., 2017; Zhang et al., 2021; Wang et al., 2021) and second-order (Elgabli et al., 2022; Li et al., 2019) methods to improve the efficacy of FL. These methods typically require access to the gradient or the Hessian of the local objective functions in their implementation to solve the optimization problem. However, using and exchanging such information raises many challenges, such as expensive communication and computation and privacy concerns (Li et al., 2020).

There's more interest recently in learning over wireless environments (Yang et al., 2020; Amiri & Gündüz, 2020; Sery & Cohen, 2020; Guo et al., 2021; Sery et al., 2021), with the increase of devices connected to servers through cellular networks. In this paper, we are interested in this scenario illustrated in Figure 1. Similarly to the aforementioned work, we are examining the case of analog communications between the devices and the server. However, it's a challenging problem as when the information is sent over the wireless channel, it becomes subject to a perturbation induced by the channel. This perturbation is not limited to additive noise, as noise is, in fact, due to thermal changes at the receiver. The channel acts as a filter for the transmitted signal (Tse & Viswanath, 2005; Björnson & Sanguinetti, 2020),

$$\hat{x} = Hx + n \tag{1}$$

where $x$ and $\hat{x} \in \mathbb{R}^d$ are the sent and received signals, respectively. $H \in \mathbb{R}^{d \times d}$ is the channel matrix, and $n \in \mathbb{R}^d$ is the additive noise, both of which are stochastic, constantly changing, and unknown. We elaborate further on the channel modeling and why we can consider it real in Appendix A for the interested reader. In federated learning, $x$ may denote the model or its gradients sent over the channel. To remove this impact, every channel element must be analyzed and removed to retrieve the sent information. This analysis is costly in computation and time resources. Thus, here our objective is to study federated learning in wireless environments without wasting resources.

Further, we're interested in exploring the potential of ZO optimization to deal with some of the difficulties demonstrated by FL. We then consider a federated learning setting where a central server coordinates with $N$ edge devices to solve an optimization problem collaboratively. The data is private to every device and the exchanges between the server and the devices is restricted to the optimization parameters. To that end, let $\mathcal{N} = \{1, ..., N\}$ be the set of devices and $\theta \in \mathbb{R}^d$ denote the global model. We define $F_i : \mathbb{R}^d \to \mathbb{R}$ as the loss function associated with the local data stored on device $i$, $\forall i \in \mathcal{N}$. The objective is to minimize the function $F : \mathbb{R}^d \to \mathbb{R}$ that is composed of the said devices' loss functions, such that

$$\min_{\theta \in \mathbb{R}^d} F(\theta) := \sum_{i=1}^{N} F_i(\theta) \quad \text{with} \quad F_i(\theta) = \mathbb{E}_{S_i \sim D_i} f_i(\theta, S_i). \tag{2}$$

$S_i$ is an i.i.d. ergodic stochastic process following a local distribution $D_i$. $S_i$ is used to model various stochastic perturbation, e.g. local data distribution among others. We further consider the case where

the devices do not have access to their gradients for computational and communication restraints, and they must estimate this gradient by querying their model only once per update. They obtain a scalar value from this query, that they must send back to the server.

## 1.1 Motivation for our work

In this subsection, we describe the various challenges in FL and how our method varies from previous work in dealing with these challenges.

**Communication bottleneck.** In general, the main idea of federated learning is that the devices receive the model from the server, make use of their data to update the gradient, and then send back their gradients without ever disclosing their data. The server then updates the model using the collected and averaged gradients, and the process repeats. Since the gradients have the same dimension as the model, in every uplink step, there are $Nd$ values that need to be uploaded, which forms a fundamental communication bottleneck in FL. To deal with this issue, some propose local multiple gradient descent steps to be done by the devices before sending their gradients back to the server to save communication resources (Khaled et al., 2020), or allow partial device participation at every iteration (Chen et al., 2018), or both (McMahan et al., 2017). Others propose lossy compression of the gradient before uploading it to the server. For example, all Konečný et al. (2016); Khirirat et al. (2018); Elgabli et al. (2020) suggest stochastic unbiased quantization approaches, where gradients are approximated with a finite set of discrete value for efficiency. Mishchenko et al. (2019) propose the quantization of gradient differences of the current and previous iterations, allowing the update to incorporate new information, while Chen et al. (2022) propose the sparsification of this difference. Sparsification means that if a vector component is not large enough, it will not be transmitted.

**Channel impact.** In federated learning over wireless channels, there's a problem with channel knowledge. When the devices upload their gradient $g \in \mathbb{R}^d$ to the server, the server receives $Hg + n$ as shown in equation (1). In Yang et al. (2020) and Fang et al. (2022) and all references within, they assume that they can remove the impact of the channel. However, as the channel matrix $H$ coefficients follow a stochastic process and there are two unknown received entities, the channel $H$ and the gradient, the knowledge of the gradient requires estimating the channel coefficients at each iteration of the FL. This requires computation resources, and more importantly, it requires resources to exchange control/reference signals between the devices and the server at each time/iteration to estimate the channel coefficients $H$. Alternatively, our work offers a much simpler approach. We don't waste resources trying to analyze the channel. We use the channel in the implementation itself. It is part of the learning. We harness it to construct our gradient estimate without the need to remove its impact, saving both computation and communication resources.

**Computation demands.** Unlike standard methods that rely on the computational capabilities of participating devices, our approach is less demanding. Devices simply receive the global model, query it with their data, and send back the scalar loss, eliminating the need for "backward pass" computation. Only the "forward pass" is performed.

**Black-box optimization in FL.** One motivation for employing ZO methods is black-box problems (Fang et al., 2022) when gradient information cannot be acquired or is complicated to compute. For example, in hyperparameter tuning, gradients cannot be calculated, as there isn't an analytic relationship between the loss function and the hyperparameters (Dai et al., 2020).

## 1.2 Challenges and contribution

Addressing nonconvexity in FL is challenging. Our ZO method must handle nonconvexity, noise, and stochasticity efficiently, which can slow down convergence in gradient techniques. Additionally, the channel's introduction adds uncertainty and constraints on the number of communication exchanges. We need to ensure consistent and reliable performance, considering unknown probability distributions and fewer function evaluations. Unlike convex cases, nonconvex optimization doesn't allow easy quantification of optimization progress. Verifying gradient convergence becomes intricate due to biased gradient estimates. Moreover, unbounded variance in one-point estimates can lead to significant gradient deviations. These challenges involve technical and intuitive complexities we navigate.

In this work, we overcome these difficulties and propose a new communication-efficient algorithm in the nonconvex setting. This algorithm differs from the standard gradient method as it entails two reception-update steps instead of one, and it's not a simple extension of FO to ZO where the devices still have to upload their full model/gradient, as is the case in Fang et al. (2022). By limiting the exchange to scalar-valued updates, we counter the communication bottleneck, and we save up to a factor of $O(d)$, in comparison to standard methods, in terms of total exchanges of variables between the devices and the server, saving a lot of execution time and allowing the convergence rate to compete with the standard FO method. We harness the nature of autocorrelated channels for truly "blind" reception of the data. We prove the convergence theoretically with one-point and two-point estimates and provide experimental evidence. An important distinction worth noting is that standard ZO methods establish convergence by focusing on the expected convergence of the exact gradient. In contrast to prior research, our approach goes further in the proof. We demonstrate the convergence of the exact gradient itself almost surely, not solely its expected value. The key element in this proof is employing Doob's martingale inequality to constrain the stochastic error resulting from estimated gradients. We finally extend the analysis to non-symmetrical channel models, i.e., channels without zero-mean, and thus provide a practical algorithm for general settings.

## 2  ALGORITHMS

This section illustrates our proposed zero-order stochastic federated learning algorithms with different gradient estimators (ZOFL).

### 2.1  THE 1P-ZOFL ALGORITHM

---

**Algorithm 1** The 1P-ZOFL algorithm

---

**Input:** Initial model $\theta_0 \in \mathbb{R}^d$, the initial step-sizes $\alpha_0$ and $\gamma_0$, and the channels' standard deviation $\sigma_h$

1: **for** $k = 0, 2, 4, ...$ **do**
2:    The server receives $\sum_{j=1}^N \frac{h_{j,k}}{\sigma_h^2} + n_{j,k}$
3:    The server broadcasts $\theta_k + \gamma_k \Phi_k \sum_{j=1}^N \left( \frac{h_{j,k}}{\sigma_h^2} + n_{j,k} \right)$ to all devices
4:    The server receives $\sum_{i=1}^N h_{i,k+1} \tilde{f}_i \left( \theta_k + \gamma_k \Phi_k \sum_{j=1}^N \left( \frac{h_{j,k}}{\sigma_h^2} + n_{j,k} \right), S_{i,k+1} \right) + n_{i,k+1}$
5:    The server multiplies the received scalar sum by $\Phi_k$ to assemble $g_k^{(1P)}$ given in (3)
6:    The server updates $\theta_{k+1} = \theta_k - \alpha_k g_k^{(1P)}$
7: **end for**

---

We consider an intermediary wireless environment between the server and each device $i$ for $i \in \mathcal{N}$ as shown in Figure 1. Wireless channels introduce a stochastic scaling on the sent signal as elaborated in equation (1). As we only send a scalar value over the channel at a time, our channel has only one scalar coefficient in addition to a scalar noise. Channel coefficients are usually autocorrelated from one timeslot to the next. Let $h_{i,k}$ denote the channel scaling affecting the sent signal from device $i$ to the server at timeslot $k$, independent from all other devices' channels. We assume $h_{i,k}$ to be a zero-mean random variable with standard deviation $\sigma_h$, $\forall i \in \mathcal{N}, \forall k \in \mathbb{N}^+$, and $n_{i,k}$ an additive noise on the transmitted signal. Assuming that the channel is time-correlated for two consecutive iterations $k$ and $k + 1$, such that the autocovariance is $\mathbb{E}[h_{i,k} h_{i,k+1}] = K_{hh}, \forall i \in \mathcal{N}, \forall k \in \mathbb{N}^+$, we present our first learning method in Algorithm 1:

The devices must carry out two communication steps. In the first, every device sends the value $\frac{1}{\sigma_h^2}$ to the server. According to equation (1), the server receives $\frac{h_{j,k}}{\sigma_h^2} + n_{j,k}$ from every device $j$. Hence, it receives the sum in step 2. Afterward, the server uses the values received to adjust the model and broadcasts it to the devices. When device $i$ receives the model, it receives $h_{i,k+1}^{DL}[\theta_k + \gamma_k \Phi_k \sum_{j=1}^N (\frac{h_{j,k}}{\sigma_h^2} + n_{j,k})] + n_{i,k+1}^{DL}$, and to simplify notation, we let the stochastic vector $[h_{i,k+1}^{DL}, n_{i,k+1}^{DL}]$ be included within the big vector $S_{i,k+1}$ of stochastic perturbations. Device $i$ then queries this received model to obtain the stochastic loss $f_i$. Then the devices send $f$ to the server in the

second communication step, and according to equation (1), the server receives the quantity indicated in step 5. Finally, the server assembles the gradient estimate and is able to update $\theta$ according to step 7. All transmissions are subject to channel scaling and additive noise. We designate them by $h$ and $n$ in the device-to-server transmission. In the server-to-devices one, we designate them by $S$. We let $\tilde{f}_i = \frac{f_i}{\sigma_h^2}$ be the normalized loss function and define $\alpha_k$ and $\gamma_k$ as two step-sizes and $\Phi_k \in \mathbb{R}^d$ as a perturbation vector generated by the server that has the same dimension as that of the model.

We emphasize here that $g_k^{(1P)}$ (in step 6) is the gradient estimate in this case, and one can see that the impact of the channel is included in the gradient estimate and hence in the learning. The major advantage of this algorithm is that each device sends only two scalar values. This is stark improvement in communication efficiency over standard federated learning algorithms that require each device to send back the whole model or local gradient of dimension $d$. In effect, it's resource draining and can be unrealistic to assume it's possible. We show in the numerical results that there is a considerable delay difference in favor of our method.

## 2.2 The 2P-ZOFL algorithm

---

**Algorithm 2** The 2P-ZOFL algorithm

---

**Input:** Initial model $\theta_0 \in \mathbb{R}^d$, the initial step-sizes $\alpha_0$ and $\gamma_0$, and the channels' standard deviation $\sigma_h$

1: **for** $k = 0, 2, 4, ...$ **do**
2:     The server receives $\sum_{j=1}^{N} \frac{h_{j,k}}{\sigma_h^2}$
3:     The server broadcasts $\theta_k + \gamma_k \Phi_k \sum_{j=1}^{N} \frac{h_{j,k}}{\sigma_h^2}$ and $\theta_k - \gamma_k \Phi_k \sum_{j=1}^{N} \frac{h_{j,k}}{\sigma_h^2}$ to all devices under the same stochastic wireless environment
4:     The server receives
$$\sum_{i=1}^{N} h_{i,k+1}\left[\tilde{f}_i\left(\theta_k + \gamma_k \Phi_k \sum_{j=1}^{N} \frac{h_{j,k}}{\sigma_h^2}, S_{i,k+1}\right) - \tilde{f}_i\left(\theta_k - \gamma_k \Phi_k \sum_{j=1}^{N} \frac{h_{j,k}}{\sigma_h^2}, S_{i,k+1}\right)\right]$$
5:     The server multiplies the received scalar sum by $\Phi_k$ to assemble $g_k^{(2P)}$ given in (4)
6:     The server updates $\theta_{k+1} = \theta_k - \alpha_k g_k^{(2P)}$
7: **end for**

---

For our second method, we aim to assemble and optimize with a two-point gradient estimate. Similarly to 1P-ZOFL, there are two communication steps. The only difference is that the server has to adjust the model twice based on the devices' feedback and broadcast the model with both adjustments. We consider that the additive noise is negligible and that the wireless environment is slowly changing. The upload communication efficiency is unaffected by the change of estimate as the functional difference is still a scalar value.

## 2.3 The estimated gradients

We provide here analysis of our ZO gradient estimates. We propose the one-point estimate:

$$g_k^{(1P)} = \Phi_k \sum_{i=1}^{N}\left[h_{i,k+1}\tilde{f}_i\left(\theta_k + \gamma_k \Phi_k \sum_{j=1}^{N}\left(\frac{h_{j,k}}{\sigma_h^2} + n_{j,k}\right), S_{i,k+1}\right) + n_{i,k+1}\right], \quad (3)$$

where $h_{i,k}$, $h_{i,k+1}$, and the noise remain unknown. This saves computation complexity and is very communication efficient as it transcends the need to send pilot signals to estimate the channel continuously. In fact, it's unrealistic to assume that the instantaneous channel can be evaluated as wireless environments typically change every $1 - 2$ ms.

In certain scenarios when the stochastic environment is changing more slowly, where the devices can query two consecutive loss functions under the same circumstances, we can use two-point estimates instead of one-point ones. In other words, whenever the server can broadcast two successive model versions under the same conditions, i.e. same $S_{i,k+1}$, our estimate can take the following form:

$$g_k^{(2P)} = \Phi_k \sum_{i=1}^{N} h_{i,k+1}\left[\tilde{f}_i\left(\theta_k + \gamma_k \Phi_k \sum_{j=1}^{N} \frac{h_{j,k}}{\sigma_h^2}, S_{i,k+1}\right) - \tilde{f}_i\left(\theta_k - \gamma_k \Phi_k \sum_{j=1}^{N} \frac{h_{j,k}}{\sigma_h^2}, S_{i,k+1}\right)\right]. \quad (4)$$

The added advantage of two-point estimates is that they increase the convergence rate as their variance w.r.t. the exact gradient is generally bounded. However, this advantage is only possible if we recognize the added noise at reception as negligible.

We next consider the following assumptions on the additive noise, the perturbation vector, and the local loss functions.

**Assumption 1** $n_{i,k}$ is assumed to be a zero-mean uncorrelated noise with bounded variance, meaning $E(n_{i,k}) = 0$ and $E(n_{i,k}^2) = \sigma_n^2 < \infty$, $\forall i \in \mathcal{N}$, $\forall k \in \mathbb{N}^+$. For any timeslot $k$, $E(n_{i,k}n_{j,k}) = 0$ if $i \neq j$. For any device $i$, $E(n_{i,k}n_{i,k'}) = 0$ if $k \neq k'$.

**Assumption 2** Let $\Phi_k = (\phi_k^1, \phi_k^2, \dots, \phi_k^d)^T$. At each iteration $k$, the server generates its $\Phi_k$ vector independently from other iterations. In addition, the elements of $\Phi_k$ are assumed i.i.d with $\mathbb{E}(\phi_k^{d_1}\phi_k^{d_2}) = 0$ for $d_1 \neq d_2$ and there exists $\alpha_2 > 0$ such that $\mathbb{E}(\phi_k^{d_j})^2 = \alpha_2$, $\forall d_j$, $\forall k$. We further assume there exists a constant $\alpha_3 > 0$ where $\|\Phi_k\| \leq \alpha_3$, $\forall k$.

**Example 1** An example of a perturbation vector satisfying Assumption 2, is picking every dimension of $\Phi_k$ from $\{-\frac{1}{\sqrt{d}}, \frac{1}{\sqrt{d}}\}$ with equal probability. Then, $\alpha_2 = \frac{1}{d}$ and $\alpha_3 = 1$.

**Assumption 3** All loss functions $\theta \mapsto f_i(\theta, S_i)$ are Lipschitz continuous with Lipschitz constant $L_{S_i}$, $|f_i(\theta, S_i) - f_i(\theta', S_i)| \leq L_{S_i}\|\theta - \theta'\|$, $\forall i \in \mathcal{N}$. In addition, $\mathbb{E}_{S_i} f_i(\theta, S_i) < \infty$, $\forall i \in \mathcal{N}$.

Let $\mathcal{H}_k = \{\theta_0, S_0, \theta_1, S_1, \dots, \theta_k, S_k\}$ denote the history sequence, then the following two Lemmas characterize our gradient estimates.

**Lemma 1** Let Assumptions 1 and 2 be satisfied and define the scalar values $c_1 = \alpha_2 \frac{K_{hh}}{\sigma_h^4}$ and $c_1' = 2c_1$, then both gradient estimators are biased w.r.t. the objective function's exact gradient $\nabla F(\theta)$. Concretely, $\mathbb{E}[g_k^{(1P)}|\mathcal{H}_k] = c_1\gamma_k(\nabla F(\theta_k) + b_k)$ and $\mathbb{E}[g_k^{(2P)}|\mathcal{H}_k] = c_1'\gamma_k(\nabla F(\theta_k) + b_k')$, $\forall k \in \mathbb{N}^+$, where $b_k$ and $b_k'$ are the bias terms.

*Proof: Refer to Appendix B.1.*

**Lemma 2** Let Assumptions 1-3 and the inequality $\|\theta_k\| < \infty$ hold almost surely. There exist two bounded constants $c_2, c_2' > 0$, such that $\mathbb{E}[\|g_k^{(1P)}\|^2|\mathcal{H}_k] \leq c_2$ and $\mathbb{E}[\|g_k^{(2P)}\|^2|\mathcal{H}_k] \leq c_2'\gamma_k^2$ almost surely.

*Proof: Refer to Appendix B.2.*

## 3 CONVERGENCE ANALYSIS

This section analyzes the behavior of our algorithms in the nonconvex setting. Assuming that a global minimizer $\theta^* \in \mathbb{R}^d$ exists such that $\min_{\theta \in \mathbb{R}^d} F(\theta) = F(\theta^*) > -\infty$ and $\nabla F(\theta^*) = 0$, we start by introducing a general necessary assumption and two estimate-specific assumptions in the subsections.

**Assumption 4** We assume the existence and the continuity of $\nabla F_i(\theta)$ and $\nabla^2 F_i(\theta)$, and that there exists a constant $\alpha_1 > 0$ such that $\|\nabla^2 F_i(\theta)\|_2 \leq \alpha_1$, $\forall i \in \mathcal{N}$.

**Lemma 3** By Assumption 4, we know that the objective function $\theta \longmapsto F(\theta)$ is L-smooth for some positive constant L, $\|\nabla F(\theta) - \nabla F(\theta')\| \leq L\|\theta - \theta'\|$, $\forall \theta, \theta' \in \mathbb{R}^d$, or equivalently, $F(\theta) \leq F(\theta') + \langle \nabla F(\theta'), \theta - \theta' \rangle + \frac{L}{2}\|\theta - \theta'\|^2$.

**Lemma 4** By Assumptions 1, 2, and 4, we can find two scalar values $c_3, c_3' > 0$ such that $\|b_k\| \leq c_3\gamma_k$ and $\|b_k'\| \leq c_3'\gamma_k$.

*Proof: Refer to Appendix B.3.*

### 3.1 1P-ZOFL CONVERGENCE

As we deal with stochastic environments, we inevitably analyze the expectation over all possible variable outcomes. From Lemma 1, we see that in expectation, our estimator deviates from the

gradient direction by the bias term. To provide that these terms don't grow larger and preferably grow smaller as the algorithms evolve, we impose that $\gamma_k$ vanishes. Additionally, to ensure that the expected norm squared of the estimator, as shown in Lemma 2, doesn't accumulate residual constant terms, we impose that the step size $\alpha_k$ vanishes. The series properties in the following assumption come from the recursive analysis of the algorithm.

**Assumption 5** *Both the step sizes $\alpha_k$ and $\gamma_k$ vanish to zero as $k \to \infty$ and the following series composed of them satisfy the convergence assumptions $\sum_{k=0}^{\infty} \alpha_k \gamma_k = \infty$, $\sum_{k=0}^{\infty} \alpha_k \gamma_k^3 < \infty$, and $\sum_{k=0}^{\infty} \alpha_k^2 < \infty$.*

**Example 2** *To satisfy Assumption 5, we consider the following form of the step sizes, $\alpha_k = \alpha_0(1 + k)^{-\upsilon_1}$ and $\gamma_k = \gamma_0(1 + k)^{-\upsilon_2}$ with $\upsilon_1, \upsilon_2 > 0$. Then, it's sufficient to find $\upsilon_1$ and $\upsilon_2$ such that $0 < \upsilon_1 + \upsilon_2 \leq 1$, $\upsilon_1 + 3\upsilon_2 > 1$, and $\upsilon_1 > 0.5$.*

We next define the stochastic error $e_k^{(1P)}$ as the difference between the value of a single realization of $g_k^{(1P)}$ and its conditional expectation given the history sequence, i.e., $e_k^{(1P)} = g_k^{(1P)} - \mathbb{E}[g_k^{(1P)}|\mathcal{H}_k]$.

The study of this noise and how it evolves is essential for the analysis of the algorithm as it gives access to the exact gradient when examining the algorithm's convergence behavior and permits us to prove that, in fact, the exact gradient converges to zero and not just the expectation of the exact gradient. This is a stronger convergence property, and it has not been done before in ZO nonconvex optimization to the best of our knowledge. The trick is to show that $e_k^{(1P)}$ is a martingale difference sequence and to apply Doob's martingale inequality to derive the following lemma.

**Lemma 5** *If all Assumptions 1-5 hold and $\|\theta_k\| < \infty$ almost surely, then for any constant $\nu > 0$, we have $\lim_{K \to \infty} \mathbb{P}(\sup_{K' \geq K} \|\sum_{k=K}^{K'} \alpha_k e_k^{(1P)}\| \geq \nu) = 0$.*

*Proof: Refer to Appendix C.1.*

The smoothness inequality allows for the first main result, leading to the second in the following theorem.

**Theorem 1** *When Assumptions 1-5 hold and given $\mathcal{H}_k$, we have $\sum_k \alpha_k \gamma_k \|\nabla F(\theta_k)\|^2 < +\infty$ and $\lim_{k \to \infty} \|\nabla F(\theta_k)\| = 0$ almost surely, meaning that the algorithm converges.*

*Proof: Refer to Appendix C.2.*

Proof sketch: We substitute the algorithm's updates in the second inequality of Lemma 3 and replace the estimate by its expectation and stochastic error. We then perform a recursive addition over the iterations $k > 0$. With Lemma 5, the conditions on the step sizes, and the upper bound estimate's squared norm, we are able to find an upper bound on $\sum_k \alpha_k \gamma_k \|\nabla F(\theta_k)\|^2$ when $k$ grows to $\infty$. The next step is to consider the hypothesis $\lim_{k \to \infty} \sup \|\nabla F(\theta_k)\| \geq \rho$, for $\rho > 0$, and prove that it contradicts with the first result.

Define $\delta_k = F(\theta_k) - F(\theta^*)$. We next find an upper bound on the convergence rate of Algorithm 1.

**Theorem 2** *Consider in addition to the assumptions in Theorem 1, that the step sizes are those of Example 2 with $\upsilon_3 = \upsilon_1 + \upsilon_2 < 1$. Then, we can write*

$$\frac{\sum_k \alpha_k \gamma_k \mathbb{E}\left[\|\nabla F(\theta_k)\|^2\right]}{\sum_k \alpha_k \gamma_k} \leq \frac{(1 - \upsilon_3)}{(K + 2)^{1-\upsilon_3} - 1}\left(\frac{2\delta_0}{c_1 \alpha_0 \gamma_0} + \frac{c_3^2 \gamma_0^2(\upsilon_1 + 3\upsilon_2)}{\upsilon_1 + 3\upsilon_2 - 1} + \frac{2c_2 L \alpha_0 \upsilon_1}{c_1 \gamma_0(2\upsilon_1 - 1)}\right). \tag{5}$$

*Proof: Refer to Appendix C.3.*

In Theorem 2, we see that the optimal choice of the exponents for the time-varying component $O(\frac{1}{K^{1-\upsilon_1-\upsilon_2}})$, is $\upsilon_1 = \frac{1}{2}$ and $\upsilon_2 = \frac{1}{6}$ for a rate of $O(\frac{1}{\sqrt[3]{K}})$. However, to prevent the constant component from growing too large, it is recommended to choose slightly larger exponents of $\upsilon_1 = \frac{1}{2} + \frac{\epsilon}{2}$ and $\upsilon_2 = \frac{1}{6} + \frac{\epsilon}{2}$, where $\epsilon$ is a small strictly positive value. This will result in a rate of $O(\frac{1}{K^{\frac{1}{3}-\epsilon}})$.

## 3.2 2P-ZOFL CONVERGENCE

Similarly to the previous subsection, we introduce an assumption regarding step sizes. The only difference comes from the fact that the upper bound on the expected norm squared of the estimate scales as $\gamma_k^2$ in Lemma 2. While $\alpha_k$ no longer needs to vanish, this does not affect the convergence rate later, so we keep the same formulation.

**Assumption 6** *Both $\alpha_k \to 0$ and $\gamma_k \to 0$ as $k \to \infty$. Besides, $\sum_{k=0}^{\infty} \alpha_k \gamma_k = \infty$, $\sum_{k=0}^{\infty} \alpha_k \gamma_k^3 < \infty$, and $\sum_{k=0}^{\infty} \alpha_k^2 \gamma_k^2 < \infty$.*

**Example 3** *Consider the same form as that in Example 2, $\alpha_k = \alpha_0(1+k)^{-\upsilon_1}$ and $\gamma_k = \gamma_0(1+k)^{-\upsilon_2}$, with $\upsilon_1, \upsilon_2 > 0$. To satisfy Assumption 6, find $\upsilon_1$ and $\upsilon_2$ such that $0 < \upsilon_1 + \upsilon_2 \leq 1$, $\upsilon_1 + 3\upsilon_2 > 1$, and $\upsilon_1 + \upsilon_2 > 0.5$.*

**Lemma 6** *Similarly, let $e_k^{(2P)} = g_k^{(2P)} - \mathbb{E}[g_k^{(2P)}|\mathcal{H}_k]$. If Assumptions 1-4 and 6 hold and $\|\theta_k\| < \infty$ almost surely, then for $\nu > 0$, we have $\lim_{K \to \infty} \mathbb{P}(\sup_{K' \geq K} \|\sum_{k=K}^{K'} \alpha_k e_k^{(2P)}\| \geq \nu) = 0$.*

*Proof: Refer to Appendix D.1.*

**Theorem 3** *Then, when Assumptions 1-4, and 6 hold, we have $\sum_k \alpha_k \gamma_k \|\nabla F(\theta_k)\|^2 < +\infty$ and $\lim_{k \to \infty} \|\nabla F(\theta_k)\| = 0$ given $\mathcal{H}_k$, almost surely, meaning that the algorithm converges.*

*Proof: Refer to Appendix D.2.*

**Theorem 4** *In addition to the assumptions of Theorem 3, let the step sizes have the form of Example 3 with $\upsilon_3 = \upsilon_1 + \upsilon_2 < 1$. Then,*

$$\frac{\sum_k \alpha_k \gamma_k \mathbb{E}\big[\|\nabla F(\theta_k)\|^2\big]}{\sum_k \alpha_k \gamma_k} \leq \frac{(1-\upsilon_3)}{(K+2)^{1-\upsilon_3}-1}\left(\frac{2\delta_0}{c_1'\alpha_0\gamma_0} + \frac{c_3'^2\gamma_0^2(\upsilon_1+3\upsilon_2)}{\upsilon_1+3\upsilon_2-1} + \frac{2c_2'L\alpha_0\gamma_0\upsilon_3}{c_1'(2\upsilon_3-1)}\right). \tag{6}$$

*Proof: Refer to Appendix D.3.*

In Theorem 4, the best exponents choice is $\upsilon_1 = \upsilon_2 = \frac{1}{4}$ which allows a rate of $O(\frac{1}{\sqrt{K}})$. To avoid the constant part growing too large, we find an arbitrarily small $\epsilon > 0$ such that $\upsilon_1 = \upsilon_2 = \frac{1}{4} + \frac{\epsilon}{2}$ for a rate of $O(\frac{1}{K^{\frac{1}{2}-\epsilon}})$.

## 3.3 NON-SYMMETRICAL CHANNELS CASE

Assuming a non-symmetrical channel model with $\mathbb{E}[h_{i,k}] = \mu_h$ and $\sigma_h^2 = \mathbb{E}[h_{i,k}^2] - \mu_h^2, \forall i, \forall k$, we provide how our gradient estimates and algorithms can be adjusted in Appendix F to account for this case. In fact, non-symmetrical channel models (e.g., Rician) offer a simplification of both analysis and implementation in comparison to symmetrical models (e.g., Rayleigh), as the non-zero mean no longer cancels out the gradient, and the design is further independent of the autocorrelation of the channels. However, with this study, we provide a generalized solution that encompasses any channel model.

## 4 EXPERIMENTAL RESULTS

For our experimental results, we ran our simulations on servers offered by our university with a Slurm workload manager. Our resources include 32 CPUs and 80GB memory over a cpu_long partition. All our codes are run in a Conda (Anaconda, 2022) virtual environment using Pytorch (Version 2.0.0) (Paszke et al., 2022) as the main library, and all datasets are accessed via Torchvision (Marcel et al., 2022). We test our algorithms in nonconvex binary image classification problems, and we compare them against the original federated learning algorithm FedAvg (McMahan et al., 2017) with exact gradient and one local update per round. However, *we don't consider the effect of the channel or any noise/stochasticity for the FedAvg algorithm*. All experiments are done for 100 devices and data batches of 10 images per user per round. Every communication round in the graphs include all steps 2 through 7 for both Algorithms 1 and 2.

For the first example, we classify photos of the two digits "0" and "1" from the MNIST dataset (LeCun & Cortes, 2005) using a nonconvex logistic regression model with a regularization parameter of 0.001. All images are divided equally among the devices and are considered to be preprocessed by being compressed to have dimension $d = 10$ using a lossy autoencoder. We run our code on 50 simulations with different random model initializations testing the accuracy in every iteration against an independent test set. The graphs in Figure 2 are averaged over all these simulations. For the non-IID data distribution, we first sort the images according to their labels and then divide them among the devices. While we can see clearly the effect of the theoretical convergence rate, both of our algorithms perform consistently well with all the different random variations influencing every simulation. Considering non-IID data distribution seems to slow down both our algorithms slightly without a major effect on the final result.

For the second example, we classify photos of "shirts" and "sneakers" from the FashionMNIST dataset (Xiao et al., 2017) using a multilayer-perceptron with an input layer of 784 units and 2 hidden layers with 200 units each using ReLu activations and a final sigmoid activation (197602 parameters). We run our code on 30 simulations with different random model initializations and average the resulting accuracy against an independent test set. The non-IID distribution is generated as in the previous example. Similarly to McMahan et al. (2017), we plot each curve by taking the best value of test-set accuracy achieved over all prior rounds. The results are shown in Figure 3. While 1P-ZOFL takes longer time to converge, 2P-ZOFL performs fairly well.

The main idea is that to converge, FedAvg requires 300 communication rounds while 2P-ZOFL requires 2000. However, by 300 rounds, each device will have uploaded $197602 \times 300 = 59280600$ scalar values to the server vs. $2000 \times 2 = 4000$. FedAvg will have 14820 more data per user. As wireless capacity is limited and there are other users using the medium, we can only send a certain amount of information per second. For a worst-case scenario where a scalar value needs one second to be uploaded, FedAvg will require around 4 more hours than 2P-ZOFL. It's true that 2P-ZOFL's convergence rate is smaller, but that doesn't mean that it is slower due to the limited capacity of wireless link as explained above. We provide a quantitative comparison with another algorithm encompassing communication-efficient strategies (local SGD and partial device participation) in Appendix E.3.

We provide all experimental details and parameter choices alongside an extra analysis of our algorithm's performance relating to its independence of the noise variance in Appendix E.

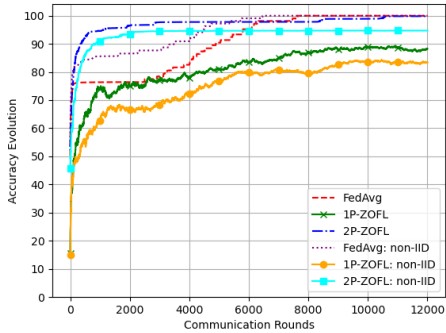
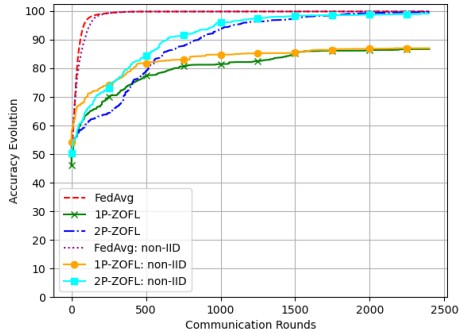

Figure 2: Accuracy evolution of 1P-ZOFL, 2P-ZOFL, and FedAvg for IID data and non-IID distribution in the logistic regression model.

Figure 3: Accuracy evolution of 1P-ZOFL, 2P-ZOFL, and FedAvg for IID and non-IID data distribution in the training example model.

## 5 CONCLUSION

This work considers a learning problem over wireless channels and proposes a new zero-order federated learning method with one-point and two-point gradient estimators. We limit the communication to scalar-valued feedback from the devices and incorporate the wireless channel in the learning algorithm. We provide theoretical and experimental evidence for convergence and find an upper bound on the convergence rate.

## REPRODUCIBILITY STATEMENT

We have made diligent efforts to enhance the reproducibility of our research findings. In the main text and the accompanying appendix, we have provided comprehensive details of our experimental procedures, data preprocessing steps, and mathematical proofs to facilitate the replication of our work. All the datasets utilized in our experiments have been cited with references to their sources, and a complete description of the data processing steps is provided in the appendix. We are committed to transparency and encourage readers to refer to the relevant sections of this paper and the appendix for a detailed account of our methodology and data to facilitate reproducibility.

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

## A  ON THE CHANNEL MODEL

As described in Section 2.2 of Chapter 2 of Tse & Viswanath (2005) and considering a double-sideband suppressed carrier amplitude modulation (DSB-SC) instead of quadrature amplitude modulation (QAM):

Having a baseband signal $x$, to send it over the channel, we modulate (multiply) it by $\sqrt{2}\cos 2\pi f_c t$ where $f_c$ is the carrier frequency and $t$ is the time index.

When sent over the channel, the transmitted signal $x$ undergoes perturbation and thus the received signal becomes:

$$z = \sqrt{2}\sum_i a_i x \cos(2\pi f_c t + \varphi_i(t)) + w(t), \tag{7}$$

where $a_i$ is the amplitude attenuation of path $i$ and $\varphi_i(t) = 2\pi f_l t + \varphi_l$ is the phase shift incurred by Doppler frequency shift $f_l$ and/or any time delay $\varphi_l$. $w(t)$ is an additive noise.

By developing the cosine term in $z$, we obtain

$$z = x\sqrt{2}\underbrace{\sum_i a_i \cos(\varphi_i(t))}_{\text{in-phase component, } I(t)}\cos(2\pi f_c t) - x\sqrt{2}\underbrace{\sum_i a_i \sin(\varphi_i(t))}_{\text{quadrature component, } Q(t)}\sin(2\pi f_c t) + w(t), \tag{8}$$

From Section 2.4.2 of Tse & Viswanath (2005): According to the central limit theorem, if there are a large number of channel paths, the in-phase and quadrature components of the received signal, which are not correlated with each other, will exhibit distributions that resemble the normal (Gaussian) distribution. Specifically, each component will have an average value of zero and a variance of $\Sigma/2$, which is equivalent to $\sigma^2$. The magnitude of the perturbation $\sqrt{I(t)^2 + Q(t)^2}$ thus becomes Rayleigh distributed. This is the Rayleigh fading model. In addition, when the line-of-sight path is large and has a known magnitude, the probabilistic model becomes a Rician fading.

Furthermore, as $I(t)$ and $Q(t)$ are orthogonal, an equivalent complex channel model $\hat{h}(t) = I(t) + jQ(t) = a(t)e^{-j\varphi(t)}$ can be derived. Since the carrier frequency $f_c$ is not involved in $\hat{h}(t)$, this representation is valid at baseband level. Thus, the complex channel model is usually used to represent the received signal $\hat{h}(t)x + n(t)$ at baseband with $\hat{h}(t)$ a complex entity.

Continuing from (8), to demodulate $z$ and obtain the baseband received signal $y$, $z$ is first multiplied by $\sqrt{2}\cos 2\pi f_c t$ then the result goes through a low pass filter.

$$
\begin{aligned}
&z\sqrt{2}\cos(2\pi f_c t)\\
=&2x\sum_i a_i \cos(\varphi_i(t))\cos^2(2\pi f_c t) - 2x\sum_i a_i \sin(\varphi_i(t))\sin(2\pi f_c t)\cos(2\pi f_c t)\\
&+ \sqrt{2}w(t)\cos(2\pi f_c t)\\
=&x\sum_i a_i \cos(\varphi_i(t))(1 + \cos(4\pi f_c t)) - x\sum_i a_i \sin(\varphi_i(t))\sin(4\pi f_c t) + \sqrt{2}w(t)\cos(2\pi f_c t)
\end{aligned}
\tag{9}
$$

After the low pass filter, we obtain the received baseband signal

$$
\begin{aligned}
y =&x\sum_i a_i \cos(\varphi_i(t)) + n(t)\\
=&\Re[\hat{h}(t)]x + n(t)\\
=&h(t)x + n(t)
\end{aligned}
\tag{10}
$$

where $n(t)$ is the baseband equivalent noise with a zero-mean Gaussian distribution and IID components (Section 2.2.4 of Tse & Viswanath (2005)) and $\Re[\hat{h}(t)] = I(t)$ is the real part of the channel.

As we are interested to send real values over the wireless channel in this paper, one can easily see how equation (10) is valid to use with a real channel $h = \Re[\hat{h}]$ following a Gaussian distribution with zero mean and variance equal to $\sigma^2$.

## B  ON THE ESTIMATED GRADIENTS

### B.1  PROOF OF LEMMA 1: BIASED ESTIMATORS

Let $g_k^{(1P)}$ have the form in (3), then the conditional expectation given $\mathcal{H}_k$ can be written as

$$
\begin{aligned}
&\mathbb{E}[g_k^{(1P)}|\mathcal{H}_k]\\
=&\mathbb{E}\left[\Phi_k\sum_{i=1}^{N}\left(h_{i,k+1}\tilde{f}_i\left(\theta_k+\gamma_k\Phi_k\sum_{j=1}^{N}\left(\frac{h_{j,k}}{\sigma_h^2}+n_{j,k}\right),S_{i,k+1}\right)+n_{i,k+1}\right)\Big|\mathcal{H}_k\right]\\
\overset{(a)}{=}&\mathbb{E}\left[\Phi_k\sum_{i=1}^{N}h_{i,k+1}\tilde{F}_i\left(\theta_k+\gamma_k\Phi_k\sum_{j=1}^{N}\left(\frac{h_{j,k}}{\sigma_h^2}+n_{j,k}\right)\right)\Big|\mathcal{H}_k\right]\\
\overset{(b)}{=}&\mathbb{E}\left[\Phi_k\left(\sum_{i=1}^{N}h_{i,k+1}\tilde{F}_i(\theta_k)+\gamma_k\sum_{i=1}^{N}h_{i,k+1}\sum_{j=1}^{N}\left(\frac{h_{j,k}}{\sigma_h^2}+n_{j,k}\right)\Phi_k^T\nabla\tilde{F}_i(\theta_k)\right.\right.\\
&\left.\left.\qquad\qquad+\gamma_k^2\sum_{i=1}^{N}h_{i,k+1}\left(\sum_{j=1}^{N}\frac{h_{j,k}}{\sigma_h^2}+n_{j,k}\right)^2\Phi_k^T\nabla^2\tilde{F}_i(\breve{\theta}_k)\Phi_k\right)\Big|\mathcal{H}_k\right]\\
\overset{(c)}{=}&\mathbb{E}\left[\Phi_k\left(\frac{\gamma_k}{\sigma_h^2}\sum_{i=1}^{N}h_{i,k+1}h_{i,k}\Phi_k^T\nabla\tilde{F}_i(\theta_k)\right.\right.\\
&\left.\left.\qquad\qquad+2\gamma_k^2 N\sum_{i=1}^{N}h_{i,k+1}\sum_{j=1}^{N}\left(\frac{h_{j,k}^2}{\sigma_h^4}+n_{j,k}^2\right)\Phi_k^T\nabla^2\tilde{F}_i(\breve{\theta}_k)\Phi_k\right)\Big|\mathcal{H}_k\right]\\
=&\frac{\gamma_k}{\sigma_h^2}\sum_{i=1}^{N}\mathbb{E}\left[h_{i,k+1}h_{i,k}\Big|\mathcal{H}_k\right]\mathbb{E}\left[\Phi_k\Phi_k^T\Big|\mathcal{H}_k\right]\nabla\tilde{F}_i(\theta_k)\\
&+2\gamma_k^2 N\sum_{i=1}^{N}\mathbb{E}\left[h_{i,k+1}\sum_{j=1}^{N}\left(\frac{h_{j,k}^2}{\sigma_h^4}+n_{j,k}^2\right)\Phi_k\Phi_k^T\nabla^2\tilde{F}_i(\breve{\theta}_k)\Phi_k\Big|\mathcal{H}_k\right]\\
\overset{(d)}{=}&\gamma_k\alpha_2\frac{K_{hh}}{\sigma_h^4}\sum_{i=1}^{N}\nabla F_i(\theta_k)+\gamma_k^2\frac{2N}{\sigma_h^2}\sum_{i=1}^{N}\mathbb{E}\left[h_{i,k+1}\sum_{j=1}^{N}\left(\frac{h_{j,k}^2}{\sigma_h^4}+n_{j,k}^2\right)\Phi_k\Phi_k^T\nabla^2\tilde{F}_i(\breve{\theta}_k)\Phi_k\Big|\mathcal{H}_k\right]\\
\overset{(e)}{=}&c_1\gamma_k(\nabla F(\theta_k)+b_k),
\end{aligned}
\tag{11}
$$

where $(a)$ is by the definition in (2) and due to Assumption 1, $(b)$ is by Taylor expansion and mean-value theorem and considering $\breve{\theta}_k$ between $\theta_k$ and $\theta_k+\gamma_k\Phi_k\sum_{j=1}^{N}\left(\frac{h_{j,k}}{\sigma_h^2}+n_{j,k}\right)$. $(c)$ is since $\mathbb{E}[h_{i,k+1}]=0$ for the first element and $\mathbb{E}[h_{i,k+1}h_{j,k}]=0$ when $i\neq j$ and the independence of noise for the second element. $(d)$ is due to Assumption 2. In $(e)$, we let $c_1=\alpha_2\frac{K_{hh}}{\sigma_h^4}$ and the bias

$$
b_k=2\gamma_k\frac{N\sigma_h^2}{\alpha_2 K_{hh}}\sum_{i=1}^{N}\mathbb{E}\left[h_{i,k+1}\sum_{j=1}^{N}\left(\frac{h_{j,k}^2}{\sigma_h^4}+n_{j,k}^2\right)\Phi_k^T\nabla^2\tilde{F}_i(\breve{\theta}_k)\Phi_k\Big|\mathcal{H}_k\right].
\tag{12}
$$

Next, let $g_k^{(2P)}$ have the form in (4), then

$$
\begin{aligned}
&\mathbb{E}[g_k^{(2P)}|\mathcal{H}_k]\\
&=\mathbb{E}\Bigg[\Phi_k\sum_{i=1}^{N}h_{i,k+1}\Big[\tilde{f}_i\Big(\theta_k+\gamma_k\Phi_k\sum_{j=1}^{N}\frac{h_{j,k}}{\sigma_h^2},S_{i,k+1}\Big)-\tilde{f}_i\Big(\theta_k-\gamma_k\Phi_k\sum_{j=1}^{N}\frac{h_{j,k}}{\sigma_h^2},S_{i,k+1}\Big)\Big]\Big|\mathcal{H}_k\Bigg]\\
&\overset{(a)}{=}\mathbb{E}\Bigg[\Phi_k\sum_{i=1}^{N}h_{i,k+1}\Big[\tilde{F}_i\Big(\theta_k+\gamma_k\Phi_k\sum_{j=1}^{N}\frac{h_{j,k}}{\sigma_h^2}\Big)-\tilde{F}_i\Big(\theta_k-\gamma_k\Phi_k\sum_{j=1}^{N}\frac{h_{j,k}}{\sigma_h^2}\Big)\Big]\Big|\mathcal{H}_k\Bigg]\\
&\overset{(b)}{=}\mathbb{E}\Bigg[\Phi_k\sum_{i=1}^{N}h_{i,k+1}\Big[\tilde{F}_i(\theta_k)+\gamma_k\sum_{j=1}^{N}\frac{h_{j,k}}{\sigma_h^2}\Phi_k^T\nabla\tilde{F}_i(\theta_k)+\gamma_k^2(\sum_{j=1}^{N}\frac{h_{j,k}}{\sigma_h^2})^2\Phi_k^T\nabla^2\tilde{F}_i(\acute{\theta}_k)\Phi_k\\
&\qquad\qquad\qquad -\Big(\tilde{F}_i(\theta_k)-\gamma_k\sum_{j=1}^{N}\frac{h_{j,k}}{\sigma_h^2}\Phi_k^T\nabla\tilde{F}_i(\theta_k)+\gamma_k^2(\sum_{j=1}^{N}\frac{h_{j,k}}{\sigma_h^2})^2\Phi_k^T\nabla^2\tilde{F}_i(\grave{\theta}_k)\Phi_k\Big)\Big]\Big|\mathcal{H}_k\Bigg]\\
&=\mathbb{E}\Bigg[\Phi_k\sum_{i=1}^{N}h_{i,k+1}\Big(2\gamma_k\sum_{j=1}^{N}\frac{h_{j,k}}{\sigma_h^2}\Phi_k^T\nabla\tilde{F}_i(\theta_k)\\
&\qquad\qquad\qquad\qquad\qquad +\gamma_k^2(\sum_{j=1}^{N}\frac{h_{j,k}}{\sigma_h^2})^2\Phi_k^T(\nabla^2\tilde{F}_i(\acute{\theta}_k)-\nabla^2\tilde{F}_i(\grave{\theta}_k))\Phi_k\Big)\Big|\mathcal{H}_k\Bigg]\\
&\overset{(c)}{=}\mathbb{E}\Bigg[\Phi_k\Big(2\frac{\gamma_k}{\sigma_h^2}\sum_{i=1}^{N}h_{i,k+1}h_{i,k}\Phi_k^T\nabla\tilde{F}_i(\theta_k)|\mathcal{H}_k\Bigg]\\
&\quad +\mathbb{E}\Bigg[\Phi_k\Big(\frac{\gamma_k^2 N}{\sigma_h^4}\sum_{i=1}^{N}h_{i,k+1}\sum_{j=1}^{N}h_{j,k}^2\Phi_k^T(\nabla^2\tilde{F}_i(\acute{\theta}_k)-\nabla^2\tilde{F}_i(\grave{\theta}_k))\Phi_k\Big)\Big|\mathcal{H}_k\Bigg]\\
&=2\frac{\gamma_k}{\sigma_h^2}\sum_{i=1}^{N}\mathbb{E}\big[h_{i,k+1}h_{i,k}\big|\mathcal{H}_k\big]\mathbb{E}\big[\Phi_k\Phi_k^T\big|\mathcal{H}_k\big]\nabla\tilde{F}_i(\theta_k)\\
&\quad +\frac{\gamma_k^2 N}{\sigma_h^4}\sum_{i=1}^{N}\mathbb{E}\Bigg[h_{i,k+1}\sum_{j=1}^{N}h_{j,k}^2\Phi_k\Phi_k^T(\nabla^2\tilde{F}_i(\acute{\theta}_k)-\nabla^2\tilde{F}_i(\grave{\theta}_k))\Phi_k\Big|\mathcal{H}_k\Bigg]\\
&\overset{(d)}{=}2\alpha_2\frac{K_{hh}}{\sigma_h^4}\gamma_k\sum_{i=1}^{N}\nabla F_i(\theta_k)+\frac{N}{\sigma_h^6}\gamma_k^2\sum_{i=1}^{N}\mathbb{E}\Bigg[h_{i,k+1}\sum_{j=1}^{N}h_{j,k}^2\Phi_k\Phi_k^T(\nabla^2 F_i(\acute{\theta}_k)-\nabla^2 F_i(\grave{\theta}_k))\Phi_k\Big|\mathcal{H}_k\Bigg]\\
&=c_1'\gamma_k(\nabla F(\theta_k)+b_k')
\end{aligned}
$$

$$(13)$$

where $(a)$ is by the definition in (2), $(b)$ is by Taylor expansion and mean-valued theorem and considering $\acute{\theta}_k$ between $\theta_k$ and $\theta_k+\gamma_k\Phi_k\sum_{j=1}^{N}\frac{h_{j,k}}{\sigma_h^2}$, and $\grave{\theta}_k$ between $\theta_k$ and $\theta_k-\gamma_k\Phi_k\sum_{j=1}^{N}\frac{h_{j,k}}{\sigma_h^2}$. $(c)$ is since $\mathbb{E}[h_{i,k+1}h_{j,k}]=0$ when $i\neq j$ in the first term. $(d)$ is due to Assumption 2. In $(e)$, we let $c_1'=2\alpha_2\frac{K_{hh}}{\sigma_h^4}$.

From (13), we can see that the estimate bias has the form

$$
b_k'=\gamma_k\frac{N}{2\alpha_2\sigma_h^2 K_{hh}}\sum_{i=1}^{N}\mathbb{E}\Bigg[h_{i,k+1}\sum_{j=1}^{N}h_{j,k}^2\Phi_k\Phi_k^T(\nabla^2 F_i(\acute{\theta}_k)-\nabla^2 F_i(\grave{\theta}_k))\Phi_k\Big|\mathcal{H}_k\Bigg]. \tag{14}
$$

## B.2 PROOF OF LEMMA 2: EXPECTED NORM SQUARED OF THE ESTIMATED GRADIENTS

Bounding the norm squared of the one-point gradient estimate,

$$
\mathbb{E}[\|g_k^{(1P)}\|^2 | \mathcal{H}_k]
$$

$$
= \mathbb{E}\left[\left\| \Phi_k \sum_{i=1}^{N} \left( h_{i,k+1} \tilde{f}_i \left( \theta_k + \gamma_k \Phi_k \sum_{j=1}^{N} \left( \frac{h_{j,k}}{\sigma_h^2} + n_{j,k} \right), S_{i,k+1} \right) + n_{i,k+1} \right) \right\|^2 \Big| \mathcal{H}_k \right]
$$

$$
= \mathbb{E}\left[ \|\Phi_k\|^2 \left( \sum_{i=1}^{N} h_{i,k+1} \tilde{f}_i \left( \theta_k + \gamma_k \Phi_k \sum_{j=1}^{N} \left( \frac{h_{j,k}}{\sigma_h^2} + n_{j,k} \right), S_{i,k+1} \right) + n_{i,k+1} \right)^2 \Big| \mathcal{H}_k \right]
$$

$$
\overset{(a)}{\leq} \alpha_3^2 N \sum_{i=1}^{N} \mathbb{E}\left[ \left( h_{i,k+1} \tilde{f}_i \left( \theta_k + \gamma_k \Phi_k \sum_{j=1}^{N} \left( \frac{h_{j,k}}{\sigma_h^2} + n_{j,k} \right), S_{i,k+1} \right) + n_{i,k+1} \right)^2 \Big| \mathcal{H}_k \right]
$$

$$
\overset{(b)}{=} \alpha_3^2 N \sum_{i=1}^{N} \mathbb{E}\left[ h_{i,k+1}^2 \tilde{f}_i^2 \left( \theta_k + \gamma_k \Phi_k \sum_{j=1}^{N} \left( \frac{h_{j,k}}{\sigma_h^2} + n_{j,k} \right), S_{i,k+1} \right) + n_{i,k+1}^2 \Big| \mathcal{H}_k \right]
$$

$$
\overset{(c)}{\leq} \alpha_3^2 N \sum_{i=1}^{N} \mathbb{E}\left[ \frac{h_{i,k+1}^2}{\sigma_h^4} \left( \|f_i(0, S_{i,k+1})\| + L_{S_{i,k+1}} \Big\| \theta_k + \gamma_k \Phi_k \sum_{j=1}^{N} \left( \frac{h_{j,k}}{\sigma_h^2} + n_{j,k} \right) \Big\| \right)^2 \Big| \mathcal{H}_k \right]
$$
$$
+ N^2 \alpha_3^2 \sigma_n^2
$$

$$
\leq \alpha_3^2 N \sum_{i=1}^{N} \mathbb{E}\left[ \frac{h_{i,k+1}^2}{\sigma_h^4} \left( \|f_i(0, S_{i,k+1})\| + L_{S_{i,k+1}} \|\theta_k\| + L_{S_{i,k+1}} \gamma_k \|\Phi_k\| \Big| \sum_{j=1}^{N} \frac{h_{j,k}}{\sigma_h^2} + n_{j,k} \Big| \right)^2 \Big| \mathcal{H}_k \right]
$$
$$
+ N^2 \alpha_3^2 \sigma_n^2
$$

$$
\overset{(d)}{\leq} 3\alpha_3^2 N \sum_{i=1}^{N} \mathbb{E}\left[ \frac{h_{i,k+1}^2}{\sigma_h^4} \left( \|f_i(0, S_{i,k+1})\|^2 + L_{S_{i,k+1}}^2 \|\theta_k\|^2 + \alpha_3^2 L_{S_{i,k+1}}^2 \gamma_k^2 \Big( \sum_{j=1}^{N} \frac{h_{j,k}}{\sigma_h^2} + n_{j,k} \Big)^2 \right) \Big| \mathcal{H}_k \right]
$$
$$
+ N^2 \alpha_3^2 \sigma_n^2
$$

$$
\overset{(e)}{\leq} \frac{3 N^2 \alpha_3^2}{\sigma_h^2} \left( \mu_S + L_S \|\theta_k\|^2 \right)
$$

$$
+ 3 N \alpha_3^4 L_S \gamma_k^2 \sum_{i=1}^{N} \mathbb{E}\left[ \frac{h_{i,k+1}^2}{\sigma_h^4} \Big( \sum_{j=1}^{N} \frac{h_{j,k}^2}{\sigma_h^4} + n_{j,k}^2 + 2 \sum_{j<l} \frac{h_{j,k} h_{l,k}}{\sigma_h^4} \Big) \Big| \mathcal{H}_k \right] + N^2 \alpha_3^2 \sigma_n^2
$$

$$
\overset{(f)}{=} \frac{3 N^2 \alpha_3^2}{\sigma_h^2} \left( \mu_S + L_S \|\theta_k\|^2 \right) + 3 N^2 \alpha_3^4 L_S \gamma_k^2 \Big( \frac{\sigma_h^4 + 2 K_{hh}^2 + (N-1)\sigma_h^4}{\sigma_h^8} + \frac{N \sigma_n^2}{\sigma_h^2} \Big) + N^2 \alpha_3^2 \sigma_n^2
$$

$$
:= c_2,
$$

$$
\tag{15}
$$

where $(a)$ is by Assumption 2 and Cauchy-Schwartz, $(b)$ is due the independence of noise, $(c)$ is by Assumption 3, and $(d)$ is again by Cauchy-Schwartz. In $(e)$, we let $\mu_S = \max_i \mathbb{E}[\|f_i(0, S_{i,k+1})\|^2 | \mathcal{H}_k]$ and $L_S = \max_i \mathbb{E}[L_{S_{i,k+1}}^2 | \mathcal{H}_k]$. $(f)$ is due to the *normally-distributed* channel random variables.

Bounding the norm squared of the two-point gradient estimate,

$$\mathbb{E}[\|g_k^{(2P)}\|^2|\mathcal{H}_k]$$

$$=\mathbb{E}\left[\left\|\Phi_k\sum_{i=1}^N h_{i,k+1}\left[\tilde{f}_i\left(\theta_k+\gamma_k\Phi_k\sum_{j=1}^N\frac{h_{j,k}}{\sigma_h^2},S_{i,k+1}\right)-\tilde{f}_i\left(\theta_k-\gamma_k\Phi_k\sum_{j=1}^N\frac{h_{j,k}}{\sigma_h^2},S_{i,k+1}\right)\right]\right\|^2\Big|\mathcal{H}_k\right]$$

$$=\mathbb{E}\left[\|\Phi_k\|^2\left(\sum_{i=1}^N h_{i,k+1}\left[\tilde{f}_i\left(\theta_k+\gamma_k\Phi_k\sum_{j=1}^N\frac{h_{j,k}}{\sigma_h^2},S_{i,k+1}\right)-\tilde{f}_i\left(\theta_k-\gamma_k\Phi_k\sum_{j=1}^N\frac{h_{j,k}}{\sigma_h^2},S_{i,k+1}\right)\right]\right)^2\Big|\mathcal{H}_k\right]$$

$$\overset{(a)}{\leq}\alpha_3^2\mathbb{E}\left[\left(\sum_{i=1}^N h_{i,k+1}\left[\tilde{f}_i\left(\theta_k+\gamma_k\Phi_k\sum_{j=1}^N\frac{h_{j,k}}{\sigma_h^2},S_{i,k+1}\right)-\tilde{f}_i\left(\theta_k-\gamma_k\Phi_k\sum_{j=1}^N\frac{h_{j,k}}{\sigma_h^2},S_{i,k+1}\right)\right]\right)^2\Big|\mathcal{H}_k\right]$$

$$\overset{(b)}{\leq}\alpha_3^2\mathbb{E}\left[\left(\sum_{i=1}^N\frac{h_{i,k+1}}{\sigma_h^2}L_{S_{i,k+1}}\left\|2\gamma_k\Phi_k\sum_{j=1}^N\frac{h_{j,k}}{\sigma_h^2}\right\|\right)^2\Big|\mathcal{H}_k\right]$$

$$\overset{(c)}{\leq}4\gamma_k^2\alpha_3^2\mathbb{E}\left[L_S\|\Phi_k\|^2\left(\sum_{i=1}^N\frac{h_{i,k+1}}{\sigma_h^2}\right)^2\left(\sum_{j=1}^N\frac{h_{j,k}}{\sigma_h^2}\right)^2\Big|\mathcal{H}_k\right]$$

$$\overset{(d)}{\leq}4\gamma_k^2\alpha_3^4N^2L_S\mathbb{E}\left[\frac{1}{\sigma_h^8}\sum_{i=1}^N h_{i,k+1}^2\sum_{j=1}^N h_{j,k}^2\Big|\mathcal{H}_k\right]$$

$$\overset{(d)}{\leq}4\gamma_k^2\alpha_3^4N^2L_S\left(\frac{1}{\sigma_h^8}\sum_{i=1}^N\mathbb{E}[h_{i,k+1}^2 h_{i,k}^2|\mathcal{H}_k]+\frac{1}{\sigma_h^8}N(N-1)\sigma_h^4\right)$$

$$\leq4\gamma_k^2\alpha_3^4N^2L_S\left(\frac{N}{\sigma_h^8}(\sigma_h^4+2K_{hh}^2)+\frac{N(N-1)}{\sigma_h^4}\right)$$

$$=\gamma_k^2\frac{4\alpha_3^4N^3L_S}{\sigma_h^4}\left(\frac{2K_{hh}^2}{\sigma_h^4}+N\right)$$

$$\overset{(e)}{=}c_2'\gamma_k^2,$$

$$(16)$$

where $(a)$ is by Assumption 2, $(b)$ is by Assumption 3, in $(c)$, $L_S=\max_i\mathbb{E}[L_{S_{i,k+1}}^2|\mathcal{H}_k]$, $(d)$ is by Cauchy-Schwartz, $(\sum_{i=1}^N x_i)^2=(\sum_{i=1}^N 1\cdot x_i)^2\leq N\sum_{i=1}^N x_i^2$, and $(d)$ is due to the *normally-distributed* channel random variables. In $(e)$, $c_2'=\frac{4\alpha_3^4N^3L_S}{\sigma_h^4}\left(\frac{2K_{hh}^2}{\sigma_h^4}+N\right)$.

### B.3 PROOF OF LEMMA 4: NORM OF THE BIAS

Considering the form of the bias in (12), by Assumptions 1, 2 and 4,

$$\|b_k\|\overset{(a)}{\leq}2\gamma_k\frac{N\sigma_h^2}{\alpha_2 K_{hh}}\sum_{i=1}^N\mathbb{E}\left[\left|h_{i,k+1}\sum_{j=1}^N\left(\frac{h_{j,k}^2}{\sigma_h^4}+n_{j,k}^2\right)\right|\|\Phi_k\|\|\Phi_k^T\|\|\nabla^2\tilde{F}_i(\breve{\theta}_k)\|\|\Phi_k\|\Big|\mathcal{H}_k\right]$$

$$\leq2\gamma_k\frac{N\alpha_1\alpha_3^3\sigma_h^2}{\alpha_2 K_{hh}}\sum_{i=1}^N\mathbb{E}\left[\left|h_{i,k+1}\sum_{j=1}^N\left(\frac{h_{j,k}^2}{\sigma_h^4}+n_{j,k}^2\right)\right|\Big|\mathcal{H}_k\right]$$

$$\overset{(b)}{\leq}2\gamma_k\frac{N\alpha_1\alpha_3^3\sigma_h^2}{\alpha_2 K_{hh}}\sum_{i=1}^N\left[\sigma_h\sqrt{\frac{2}{\pi}}\left(2K_{hh}+\sqrt{\sigma_h^4-K_{hh}^2}\right)+(N-1)\sigma_h^3\sqrt{\frac{2}{\pi}}+N\sqrt{\frac{2}{\pi}}\sigma_h\sigma_n^2\right]$$

$$=2\gamma_k\frac{N^2\alpha_1\alpha_3^3\sigma_h^3}{\alpha_2 K_{hh}}\sqrt{\frac{2}{\pi}}\left[\left(2K_{hh}+\sqrt{\sigma_h^4-K_{hh}^2}\right)+(N-1)\sigma_h^2+N\sigma_n^2\right]$$

$$\overset{(c)}{=}c_3\gamma_k,$$

$$(17)$$

where $(a)$ is due to Jensen's inequality, $(b)$ is by using the half-normal distribution for normal random variables in absolute value explained in the following paragraph, and in $(c)$, $c_3 = 2\frac{N^2\alpha_1\alpha_3^3\sigma_h^3}{\alpha_2 K_{hh}}\sqrt{\frac{2}{\pi}}\left[\left(2K_{hh} + \sqrt{\sigma_h^4 - K_{hh}^2}\right) + (N-1)\sigma_h^2 + N\sigma_n^2\right]$.

Let $X$ and $Y$ be two variables representing time-correlated channel realizations at times $k$ and $k'$ respectively. Assume they follow the $\mathcal{N}(0,\sigma)$ distribution and they have a correlation coefficient $\varrho$. Then, we can write $Y = \varrho X + \sqrt{1 - \varrho^2}Z$, where $Z$ is independent of X and following the same distribution $\mathcal{N}(0,\sigma)$. Then,

$$
\begin{aligned}
\mathbb{E}[|YX^2|] &= \mathbb{E}[|(\varrho X + \sqrt{1-\varrho^2}Z)X^2|] \\
&= \mathbb{E}[|\varrho X^3 + \sqrt{1-\varrho^2}ZX^2|] \\
&\leq \mathbb{E}[\varrho|X^3| + \sqrt{1-\varrho^2}|ZX^2|] \\
&= 2\varrho\sqrt{\frac{2}{\pi}}\sigma^3 + \sqrt{1-\varrho^2}\sqrt{\frac{2}{\pi}}\sigma \times \sigma^2 \\
&= (2\varrho + \sqrt{1-\varrho^2})\sqrt{\frac{2}{\pi}}\sigma^3.
\end{aligned}
$$

If we substitute $\sigma = \sigma_h$ and $\varrho = \frac{K_{hh}}{\sigma_h^2}$, we obtain the previous inequality $(b)$.

Similarly, the bias of (14) can be bounded from above using Assumptions 2 and 4, as

$$
\begin{aligned}
\|b_k'\| &\leq \gamma_k \frac{N}{2\alpha_2\sigma_h^2 K_{hh}}\sum_{i=1}^{N}\mathbb{E}\left[\left|h_{i,k+1}\sum_{j=1}^{N}h_{j,k}^2\right|\|\Phi_k\|\|\Phi_k^T\|\|\nabla^2 F_i(\acute{\theta}_k) - \nabla^2 F_i(\grave{\theta}_k)\|\|\Phi_k\|\Big|\mathcal{H}_k\right] \\
&\leq 2\gamma_k\alpha_1\alpha_3^3\frac{N}{2\alpha_2\sigma_h^2 K_{hh}}\sum_{i=1}^{N}\mathbb{E}\left[\left|h_{i,k+1}(h_{i,k}^2 + \sum_{j\neq i}h_{j,k}^2)\right|\Big|\mathcal{H}_k\right] \\
&\leq 2\gamma_k\alpha_1\alpha_3^3\frac{N}{2\alpha_2\sigma_h^2 K_{hh}}\sum_{i=1}^{N}\left[\sigma_h\sqrt{\frac{2}{\pi}}\left(2K_{hh} + \sqrt{\sigma_h^4 - K_{hh}^2}\right) + (N-1)\sigma_h^3\sqrt{\frac{2}{\pi}}\right] \\
&\leq \gamma_k\alpha_1\alpha_3^3\frac{N^2}{\alpha_2\sigma_h K_{hh}}\sqrt{\frac{2}{\pi}}\left[2K_{hh} + \sqrt{\sigma_h^4 - K_{hh}^2} + (N-1)\sigma_h^2\right] \\
&= c_3'\gamma_k,
\end{aligned}
$$

$$(18)$$

with $c_3' = \alpha_1\alpha_3^3\frac{N^2}{\alpha_2\sigma_h K_{hh}}\sqrt{\frac{2}{\pi}}\left[2K_{hh} + \sqrt{\sigma_h^4 - K_{hh}^2} + (N-1)\sigma_h^2\right]$.

## C    1P-ZOFL ALGORITHM CONVERGENCE

### C.1    STOCHASTIC NOISE

To prove Lemma 5, we begin by demonstrating that the sequence $\{\sum_{k=K}^{K'}\alpha_k e_k^{(1P)}\}_{K'\geq K}$ is a martingale. Since $g_k^{(1P)}$ and $g_{k'}^{(1P)}$ are independent if $k \neq k'$ and

$$
\begin{aligned}
\mathbb{E}[e_k^{(1P)}] &= \mathbb{E}[g_k^{(1P)} - \mathbb{E}[g_k^{(1P)}|\mathcal{H}_k]] \\
&= \mathbb{E}_{\mathcal{H}_k}\left[\mathbb{E}\left[g_k^{(1P)} - \mathbb{E}[g_k^{(1P)}|\mathcal{H}_k]\Big|\mathcal{H}_k\right]\right] \\
&= 0
\end{aligned}
$$

by the law of total expectation, the sequence is a martingale. Therefore, for any constant $\nu > 0$, we can state

$$
\begin{aligned}
\mathbb{P}(\sup_{K' \geq K} \| \sum_{k=K}^{K'} \alpha_k e_k^{(1P)} \| \geq \nu) &\overset{(a)}{\leq} \mathbb{E}(\| \sum_{k=K}^{K'} \alpha_k e_k^{(1P)} \|^2) \\
&= \frac{1}{\nu^2} \mathbb{E}(\sum_{k=K}^{K'} \sum_{k'=K}^{K'} \alpha_k \alpha_{k'} \langle e_k^{(1P)}, e_{k'}^{(1P)} \rangle) \\
&\overset{(b)}{=} \frac{1}{\nu^2} \mathbb{E}(\sum_{k=K}^{K'} \| \alpha_k e_k^{(1P)} \|^2) \\
&\leq \frac{1}{\nu^2} \sum_{k=K}^{\infty} \mathbb{E}(\alpha_k^2 \| g_k^{(1P)} - \mathbb{E}[g_k^{(1P)} | \mathcal{H}_k] \|^2) \\
&= \frac{1}{\nu^2} \sum_{k=K}^{\infty} \alpha_k^2 \mathbb{E}(\| g_k^{(1P)} \|^2) - \mathbb{E}_{\mathcal{H}_k}(\| \mathbb{E}[g_k^{(1P)} | \mathcal{H}_k] \|^2) \\
&\leq \frac{1}{\nu^2} \sum_{k=K}^{\infty} \alpha_k^2 \mathbb{E}(\| g_k^{(1P)} \|^2) \\
&\overset{(c)}{\leq} \frac{c_2}{\nu^2} \sum_{k=K}^{\infty} \alpha_k^2,
\end{aligned}
$$

where $(a)$ is due to Doob's martingale inequality (Doob, 1953), $(b)$ is since $\mathbb{E}[\langle e_k^{(1P)}, e_{k'}^{(1P)} \rangle] = 0$ for any $k \neq k'$, and $(c)$ is by Lemma 2.

Since $c_2$ is a bounded constant and $\lim_{K \to \infty} \sum_{k=K}^{\infty} \alpha_k^2 = 0$ by Assumption 5, we get $\lim_{K \to \infty} \frac{c_2}{\nu^2} \sum_{k=K}^{\infty} \alpha_k^2 = 0$ for any bounded constant $\nu$. Hence, the probability that $\| \sum_{k=K}^{K'} \alpha_k e_k^{(1P)} \| \geq \nu$ also vanishes as $K \to \infty$, which concludes the proof.

## C.2 PROOF OF THEOREM 1: CONVERGENCE ANALYSIS

By the $L$-smoothness assumption and the algorithm update step $\theta_{k+1} = \theta_k - \alpha_k g_k^{(1P)}$, we have

$$
\begin{aligned}
F(\theta_{k+1}) &\leq F(\theta_k) - \alpha_k \langle \nabla F(\theta_k), g_k^{(1P)} \rangle + \frac{\alpha_k^2 L}{2} \| g_k^{(1P)} \|^2 \\
&= F(\theta_k) - \alpha_k \langle \nabla F(\theta_k), g_k^{(1P)} - \mathbb{E}[g_k^{(1P)} | \mathcal{H}_k] + \mathbb{E}[g_k^{(1P)} | \mathcal{H}_k] \rangle + \frac{\alpha_k^2 L}{2} \| g_k^{(1P)} \|^2 \\
&= F(\theta_k) - \alpha_k \langle \nabla F(\theta_k), e_k^{(1P)} \rangle - c_1 \alpha_k \gamma_k \langle \nabla F(\theta_k), \nabla F(\theta_k) + b_k \rangle + \frac{\alpha_k^2 L}{2} \| g_k^{(1P)} \|^2 \\
&= F(\theta_k) - \alpha_k \langle \nabla F(\theta_k), e_k^{(1P)} \rangle - c_1 \alpha_k \gamma_k \| \nabla F(\theta_k) \|^2 - c_1 \alpha_k \gamma_k \langle \nabla F(\theta_k), b_k \rangle \\
&\quad + \frac{\alpha_k^2 L}{2} \| g_k^{(1P)} \|^2 \\
&\overset{(a)}{\leq} F(\theta_k) - \alpha_k \langle \nabla F(\theta_k), e_k^{(1P)} \rangle - c_1 \alpha_k \gamma_k \| \nabla F(\theta_k) \|^2 + \frac{c_1 \alpha_k \gamma_k}{2} \| \nabla F(\theta_k) \|^2 \\
&\quad + \frac{c_1 \alpha_k \gamma_k}{2} \| b_k \|^2 + \frac{\alpha_k^2 L}{2} \| g_k^{(1P)} \|^2 \\
&= F(\theta_k) - \alpha_k \langle \nabla F(\theta_k), e_k^{(1P)} \rangle - \frac{c_1 \alpha_k \gamma_k}{2} \| \nabla F(\theta_k) \|^2 + \frac{c_1 \alpha_k \gamma_k}{2} \| b_k \|^2 + \frac{\alpha_k^2 L}{2} \| g_k^{(1P)} \|^2
\end{aligned}
$$
$$(19)$$

where $(a)$ is by $-\langle a, b \rangle \leq \frac{1}{2} \| a \|^2 + \frac{1}{2} \| b \|^2$.

By taking the telescoping sum, we get

$$F(\theta^*) \leq F(\theta_{K+1}) \leq F(\theta_0) - \frac{c_1}{2} \sum_{k=0}^{K} \alpha_k \gamma_k \|\nabla F(\theta_k)\|^2 - \sum_{k=0}^{K} \alpha_k \langle \nabla F(\theta_k), e_k^{(1P)} \rangle$$
$$+ \frac{c_1}{2} \sum_{k=0}^{K} \alpha_k \gamma_k \|b_k\|^2 + \frac{c_2 L}{2} \sum_{k=0}^{K} \alpha_k^2 \|g_k^{(1P)}\|^2 \tag{20}$$

Hence,

$$\sum_{k=0}^{K} \alpha_k \gamma_k \|\nabla F(\theta_k)\|^2 \leq \frac{2}{c_1}(F(\theta_0) - F(\theta^*)) - \frac{2}{c_1} \sum_{k=0}^{K} \alpha_k \langle \nabla F(\theta_k), e_k^{(1P)} \rangle + \sum_{k=0}^{K} \alpha_k \gamma_k \|b_k\|^2$$
$$+ \frac{c_2 L}{c_1} \sum_{k=0}^{K} \alpha_k^2 \|g_k^{(1P)}\|^2 \tag{21}$$

By Assumption 3, $\|\nabla F(\theta_k)\|$ is bounded for any $\theta_k \in \mathbb{R}^d$ and by Lemma 5, we have

$$\lim_{K \to \infty} \| \sum_{k=0}^{K} \alpha_k \langle \nabla F(\theta_k), e_k^{(1P)} \rangle \| < \infty. \tag{22}$$

From Lemma 4, we know that $\|b_k\|^2 \sim \gamma_k^2$. Hence, by Assumption 5,

$$\lim_{K \to \infty} \sum_{k=0}^{K} \alpha_k \gamma_k^3 < \infty. \tag{23}$$

From Lemma 2 and by looking closely at the use of the Lipschitz continuity property in (15), we can say $\|g_k^{(1P)}\|^2 \leq c$ for some $c > 0$. Thus, again by Assumption 5,

$$\lim_{K \to \infty} \sum_{k=0}^{K} \alpha_k^2 < \infty. \tag{24}$$

We conclude that

$$\lim_{K \to \infty} \sum_{k=0}^{K} \alpha_k \gamma_k \|\nabla F(\theta_k)\|^2 < \infty. \tag{25}$$

Moreover, since the series $\sum_k \alpha_k \gamma_k$ diverges by Assumption 5, we have

$$\lim_{k \to \infty} \inf \|\nabla F(\theta_k)\| = 0. \tag{26}$$

To prove that $\lim_{k \to \infty} \|\nabla F(\theta_k)\| = 0$, we consider the hypothesis H) $\lim_{k \to \infty} \sup \|\nabla F(\theta_k)\| \geq \rho$ for an arbitrary $\rho > 0$.

Assume (H) to be true. Then, we can always find an arbitrary subsequence $\left( \|\nabla F(\theta_{k_l})\| \right)_{l \in \mathbb{N}}$ of $\|\nabla F(\theta_k)\|$, such that $\|\nabla F(\theta_{k_l})\| \geq \rho - \varepsilon, \forall l$, for $\rho - \varepsilon > 0$ and $\varepsilon > 0$.

Then, by the $L$-smoothness property and applying the descent step of the algorithm,

$$\|\nabla F(\theta_{k_l+1})\| \geq \|\nabla F(\theta_{k_l})\| - \|\nabla F(\theta_{k_l+1}) - \nabla F(\theta_{k_l})\|$$
$$\geq \rho - \varepsilon - L\|\theta_{k_l+1} - \theta_{k_l}\|$$
$$= \rho - \varepsilon - L\alpha_{k_l}\|g_{k_l}\|$$
$$\geq \rho - \varepsilon - L\sqrt{c}\alpha_{k_l}, \tag{27}$$

Since $k_l \to \infty$ as $l \to \infty$, we can always find a subsequence of $(k_{l_p})_{p \in \mathbb{N}}$ such that $k_{l_{p+1}} - k_{l_p} > 1$. As $\alpha_{k_l}$ is vanishing, we consider $(k_l)_{l \in \mathbb{N}}$ starting from $\alpha_{k_l} < \frac{\rho - \varepsilon}{L\sqrt{c}}$. Thus,

$$
\begin{aligned}
\sum_{k=0}^{\infty} & \alpha_{k+1}\gamma_{k+1} \|\nabla F(\theta_{k+1})\|^2 \\
&\geq (\rho - \varepsilon)^2 \sum_{k=0}^{\infty} \alpha_{k+1}\gamma_{k+1} - 2(\rho - \varepsilon)L\sqrt{c} \sum_{k=0}^{\infty} \alpha_{k+1}\gamma_{k+1}\alpha_k + L^2 c \sum_{k=0}^{\infty} \alpha_{k+1}\gamma_{k+1}\alpha_k^2 \\
&\geq (\rho - \varepsilon)^2 \sum_{k=0}^{\infty} \alpha_{k+1}\gamma_{k+1} - 2(\rho - \varepsilon)L\sqrt{c} \sum_{k=0}^{\infty} \alpha_k^2\gamma_{k+1} + L^2 c \sum_{k=0}^{\infty} \alpha_{k+1}\gamma_{k+1}\alpha_k^2 \\
&= +\infty,
\end{aligned}
\tag{28}
$$

as the first series diverges, and the second and the third converge by Assumption 5. This implies that the series $\sum_k \alpha_k\gamma_k \|\nabla F(\theta_k)\|^2$ diverges. This is a contradiction as this series converges almost surely by (25). Therefore, hypothesis (H) cannot be true and $\|\nabla F(\theta_k)\|$ converges to zero almost surely.

### C.3  PROOF OF THEOREM 2: CONVERGENCE RATE

Starting again from the $L$-smoothness in Lemma 3 and the algorithm update step $\theta_{k+1} = \theta_k - \alpha_k g_k^{(1P)}$, we have

$$
F(\theta_{k+1}) \leq F(\theta_k) - \alpha_k \langle \nabla F(\theta_k), g_k^{(1P)} \rangle + \frac{\alpha_k^2 L}{2} \|g_k^{(1P)}\|^2.
\tag{29}
$$

Taking the conditional expectation given $\mathcal{H}_k$,

$$
\begin{aligned}
F(\theta_{k+1}) &\leq F(\theta_k) - c_1\alpha_k\gamma_k \langle \nabla F(\theta_k), \nabla F(\theta_k) + b_k \rangle + \frac{\alpha_k^2 L c_2}{2} \\
&= F(\theta_k) - c_1\alpha_k\gamma_k \|\nabla F(\theta_k)\|^2 - c_1\alpha_k\gamma_k \langle \nabla F(\theta_k), b_k \rangle + \frac{\alpha_k^2 L c_2}{2} \\
&\overset{(a)}{\leq} F(\theta_k) - c_1\alpha_k\gamma_k \|\nabla F(\theta_k)\|^2 + \frac{c_1\alpha_k\gamma_k}{2} \|\nabla F(\theta_k)\|^2 + \frac{c_1\alpha_k\gamma_k}{2} \|b_k\|^2 + \frac{\alpha_k^2 L c_2}{2} \\
&= F(\theta_k) - \frac{c_1\alpha_k\gamma_k}{2} \|\nabla F(\theta_k)\|^2 + \frac{c_1\alpha_k\gamma_k}{2} \|b_k\|^2 + \frac{\alpha_k^2 L c_2}{2}
\end{aligned}
\tag{30}
$$

where $(a)$ is by $-\langle a, b \rangle \leq \frac{1}{2}\|a\|^2 + \frac{1}{2}\|b\|^2$

By considering a large value $K > 0$ and taking the telescoping sum of (30), we get

$$
\mathbb{E}[F(\theta_{K+1})|\mathcal{H}_K] \leq F(\theta_0) - \frac{c_1}{2} \sum_{k=0}^{K} \alpha_k\gamma_k \|\nabla F(\theta_k)\|^2 + \frac{c_1}{2} \sum_{k=0}^{K} \alpha_k\gamma_k \|b_k\|^2 + \frac{L c_2}{2} \sum_{k=0}^{K} \alpha_k^2
\tag{31}
$$

Given the assumption that $F(\theta^*) = \min_{\theta \in \mathbb{R}^d} F(\theta)$ exists, we know that $\delta_k = F(\theta_k) - F(\theta^*) \geq 0$. Then,

$$
0 \leq \mathbb{E}[\delta_{K+1}|\mathcal{H}_K] \leq \delta_0 - \frac{c_1}{2} \sum_{k=0}^{K} \alpha_k\gamma_k \|\nabla F(\theta_k)\|^2 + \frac{c_1 c_3}{2} \sum_{k=0}^{K} \alpha_k\gamma_k^3 + \frac{L c_2}{2} \sum_{k=0}^{K} \alpha_k^2
\tag{32}
$$

Finally,

$$
\sum_k \alpha_k\gamma_k \mathbb{E}[\|\nabla F(\theta_k)\|^2] \leq \frac{2}{c_1 c_3}\delta_0 + \sum_k \alpha_k\gamma_k^3 + \frac{L c_2}{c_1} \sum_k \alpha_k^2
\tag{33}
$$

Let $\alpha_k$ and $\gamma_k$ have the forms given in Example 2. We know that, $\forall K > 0$,

$$
\begin{aligned}
\sum_{k=0}^{K} \alpha_k \gamma_k^3 = \alpha_0 \gamma_0^3 + \sum_{k=1}^{K} \alpha_k \gamma_k^3 &\leq \alpha_0 \gamma_0^3 \left( 1 + \int_0^K (x+1)^{-\upsilon_1 - 3\upsilon_2} dx \right) \\
&= \alpha_0 \gamma_0^3 \left( 1 + \frac{1}{\upsilon_1 + 3\upsilon_2 - 1} - \frac{(K+1)^{-\upsilon_1 - 3\upsilon_2 + 1}}{\upsilon_1 + 3\upsilon_2 - 1} \right) \\
&\leq \alpha_0 \gamma_0^3 \left( 1 + \frac{1}{\upsilon_1 + 3\upsilon_2 - 1} \right) \\
&= \alpha_0 \gamma_0^3 \left( \frac{\upsilon_1 + 3\upsilon_2}{\upsilon_1 + 3\upsilon_2 - 1} \right).
\end{aligned}
\tag{34}
$$

Similarly,

$$
\sum_{k=0}^{K} \alpha_k^2 \leq \alpha_0^2 \left( \frac{2\upsilon_1}{2\upsilon_1 - 1} \right)
\tag{35}
$$

- Next, when $0 < \upsilon_1 + \upsilon_2 < 1$,

$$
\begin{aligned}
\sum_{k=0}^{K} \alpha_k \gamma_k &\geq \alpha_0 \gamma_0 \int_0^{K+1} (x+1)^{-\upsilon_1 - \upsilon_2} dx \\
&= \frac{\alpha_0 \gamma_0}{(1 - \upsilon_1 - \upsilon_2)} \left( (K+2)^{1 - \upsilon_1 - \upsilon_2} - 1 \right).
\end{aligned}
\tag{36}
$$

Thus, making use of inequality (33)

$$
\begin{aligned}
\frac{\sum_k \alpha_k \gamma_k \mathbb{E}[\|\nabla F(\theta_k)\|^2]}{\sum_k \alpha_k \gamma_k} \leq &\frac{(1 - \upsilon_1 - \upsilon_2)}{(K+2)^{1 - \upsilon_1 - \upsilon_2} - 1} \\
&\times \left( \frac{2}{c_1 \alpha_0 \gamma_0} \delta_0 + c_3^2 \gamma_0^2 \left( \frac{\upsilon_1 + 3\upsilon_2}{\upsilon_1 + 3\upsilon_2 - 1} \right) + \frac{L c_2 \alpha_0}{c_1 \gamma_0} \left( \frac{2\upsilon_1}{2\upsilon_1 - 1} \right) \right)
\end{aligned}
\tag{37}
$$

- Otherwise, when $\upsilon_1 + \upsilon_2 = 1$,

$$
\begin{aligned}
\sum_{k=0}^{K} \alpha_k \gamma_k &\geq \alpha_0 \gamma_0 \int_0^{K+1} \frac{1}{x+1} dx \\
&= \alpha_0 \gamma_0 \ln(K+2).
\end{aligned}
\tag{38}
$$

Thus, we get

$$
\begin{aligned}
\frac{\sum_k \alpha_k \gamma_k \mathbb{E}[\|\nabla F(\theta_k)\|^2]}{\sum_k \alpha_k \gamma_k} \leq &\frac{1}{\ln(K+2)} \\
&\times \left( \frac{2}{c_1 \alpha_0 \gamma_0} \delta_0 + c_3^2 \gamma_0^2 \left( \frac{\upsilon_1 + 3\upsilon_2}{\upsilon_1 + 3\upsilon_2 - 1} \right) + \frac{L c_2 \alpha_0}{c_1 \gamma_0} \left( \frac{2\upsilon_1}{2\upsilon_1 - 1} \right) \right)
\end{aligned}
\tag{39}
$$

## D  2P-ZOFL ALGORITHM CONVERGENCE

### D.1  STOCHASTIC NOISE

Since $g_k^{(2P)}$ and $g_{k'}^{(2P)}$ are independent if $k \neq k'$ and

$$
\begin{aligned}
\mathbb{E}[e_k^{(2P)}] &= \mathbb{E}[g_k^{(2P)} - \mathbb{E}[g_k^{(2P)} | \mathcal{H}_k]] \\
&= \mathbb{E}_{\mathcal{H}_k} \left[ \mathbb{E}\left[ g_k^{(2P)} - \mathbb{E}[g_k^{(2P)} | \mathcal{H}_k] \big| \mathcal{H}_k \right] \right], \\
&= 0
\end{aligned}
$$

the sequence is a martingale. Thus, for any $\nu > 0$,

$$\mathbb{P}(\sup_{K' \geq K} \| \sum_{k=K}^{K'} \alpha_k e_k^{(2P)}\| \geq \nu) \overset{(a)}{\leq} \mathbb{E}(\| \sum_{k=K}^{K'} \alpha_k e_k^{(2P)}\|^2)$$

$$= \frac{1}{\nu^2} \mathbb{E}(\sum_{k=K}^{K'} \sum_{k'=K}^{K'} \alpha_k \alpha_{k'} \langle e_k^{(2P)}, e_{k'}^{(2P)} \rangle)$$

$$\overset{(b)}{=} \frac{1}{\nu^2} \mathbb{E}(\sum_{k=K}^{K'} \| \alpha_k e_k^{(2P)}\|^2)$$

$$\leq \frac{1}{\nu^2} \sum_{k=K}^{\infty} \mathbb{E}(\alpha_k^2 \| g_k^{(2P)} - \mathbb{E}[g_k^{(2P)}|\mathcal{H}_k]\|^2)$$

$$= \frac{1}{\nu^2} \sum_{k=K}^{\infty} \alpha_k^2 \mathbb{E}(\| g_k^{(2P)}\|^2) - \mathbb{E}_{\mathcal{H}_k}(\| \mathbb{E}[g_k^{(2P)}|\mathcal{H}_k]\|^2)$$

$$\leq \frac{1}{\nu^2} \sum_{k=K}^{\infty} \alpha_k^2 \mathbb{E}(\| g_k^{(2P)}\|^2)$$

$$\overset{(c)}{\leq} \frac{c_2'}{\nu^2} \sum_{k=K}^{\infty} \alpha_k^2 \gamma_k^2,$$

where $(a)$ is due to Doob's martingale inequality (Doob, 1953), $(b)$ is since $\mathbb{E}[\langle e_k^{(2P)}, e_{k'}^{(2P)} \rangle] = 0$ for any $k \neq k'$, and $(c)$ is by Lemma 2.

Since $c_2'$ is a bounded constant and $\lim_{K \to \infty} \sum_{k=K}^{\infty} \alpha_k^2 \gamma_k^2 = 0$ by Assumption 6, we get $\lim_{K \to \infty} \frac{c_2'}{\nu^2} \sum_{k=K}^{\infty} \alpha_k^2 \gamma_k^2 = 0$ for any bounded constant $\nu$. Hence, the probability that $\| \sum_{k=K}^{K'} \alpha_k e_k^{(2P)}\| \geq \nu$ also vanishes as $K \to \infty$, which concludes the proof.

### D.2  PROOF OF THEOREM 3: CONVERGENCE ANALYSIS

By the $L$-smoothness assumption, we have

$$F(\theta_{k+1}) \leq F(\theta_k) - \alpha_k \langle \nabla F(\theta_k), g_k^{(2P)} \rangle + \frac{\alpha_k^2 L}{2} \| g_k^{(2P)}\|^2$$

$$= F(\theta_k) - \alpha_k \langle \nabla F(\theta_k), g_k^{(2P)} - \mathbb{E}[g_k^{(2P)}|\mathcal{H}_k] + \mathbb{E}[g_k^{(2P)}|\mathcal{H}_k] \rangle + \frac{\alpha_k^2 L}{2} \| g_k^{(2P)}\|^2$$

$$= F(\theta_k) - \alpha_k \langle \nabla F(\theta_k), e_k^{(2P)} \rangle - c_1' \alpha_k \gamma_k \langle \nabla F(\theta_k), \nabla F(\theta_k) + b_k' \rangle + \frac{\alpha_k^2 L}{2} \| g_k^{(2P)}\|^2$$

$$= F(\theta_k) - \alpha_k \langle \nabla F(\theta_k), e_k^{(2P)} \rangle - c_1' \alpha_k \gamma_k \| \nabla F(\theta_k)\|^2 - c_1' \alpha_k \gamma_k \langle \nabla F(\theta_k), b_k' \rangle + \frac{\alpha_k^2 L}{2} \| g_k^{(2P)}\|^2$$

$$\overset{(a)}{\leq} F(\theta_k) - \alpha_k \langle \nabla F(\theta_k), e_k^{(2P)} \rangle - c_1' \alpha_k \gamma_k \| \nabla F(\theta_k)\|^2 + \frac{c_1' \alpha_k \gamma_k}{2} \| \nabla F(\theta_k)\|^2$$

$$+ \frac{c_1' \alpha_k \gamma_k}{2} \| b_k'\|^2 + \frac{\alpha_k^2 L}{2} \| g_k^{(2P)}\|^2$$

$$= F(\theta_k) - \alpha_k \langle \nabla F(\theta_k), e_k^{(2P)} \rangle - \frac{c_1' \alpha_k \gamma_k}{2} \| \nabla F(\theta_k)\|^2 + \frac{c_1' \alpha_k \gamma_k}{2} \| b_k'\|^2 + \frac{\alpha_k^2 L}{2} \| g_k^{(2P)}\|^2$$

$$(40)$$

where $(a)$ is by $-\langle a, b \rangle \leq \frac{1}{2}\|a\|^2 + \frac{1}{2}\|b\|^2$.

By taking the telescoping sum, we get

$$
\begin{aligned}
F(\theta^*) \le F(\theta_{K+1}) \le F(\theta_0) &- \frac{c_1'}{2} \sum_{k=0}^{K} \alpha_k \gamma_k \|\nabla F(\theta_k)\|^2 - \sum_{k=0}^{K} \alpha_k \langle \nabla F(\theta_k), e_k^{(2P)} \rangle \\
&+ \frac{c_1'}{2} \sum_{k=0}^{K} \alpha_k \gamma_k \|b_k'\|^2 + \frac{c_2' L}{2} \sum_{k=0}^{K} \alpha_k^2 \|g_k^{(2P)}\|^2
\end{aligned}
\tag{41}
$$

Hence,

$$
\begin{aligned}
\sum_{k=0}^{K} \alpha_k \gamma_k \|\nabla F(\theta_k)\|^2 \le &\frac{2}{c_1'} (F(\theta_0) - F(\theta^*)) - \frac{2}{c_1'} \sum_{k=0}^{K} \alpha_k \langle \nabla F(\theta_k), e_k^{(2P)} \rangle + \sum_{k=0}^{K} \alpha_k \gamma_k \|b_k'\|^2 \\
&+ \frac{c_2' L}{c_1'} \sum_{k=0}^{K} \alpha_k^2 \|g_k^{(2P)}\|^2
\end{aligned}
\tag{42}
$$

By Assumption 3, $\|\nabla F(\theta_k)\|$ is bounded for any $\theta_k \in \mathbb{R}^d$ and by Lemma 6, we have

$$
\lim_{K \to \infty} \| \sum_{k=0}^{K} \alpha_k \langle \nabla F(\theta_k), e_k \rangle \| < \infty.
\tag{43}
$$

From Lemma 4, we know that $\|b_k'\|^2 \sim \gamma_k^2$. Hence, by Assumption 6,

$$
\lim_{K \to \infty} \sum_{k=0}^{K} \alpha_k \gamma_k^3 < \infty.
\tag{44}
$$

From Lemma 2 and by looking closely at the use of the Lipschitz continuity property in (16), we can say $\|g_k^{(2P)}\|^2 \le c' \gamma_k^2$ for some $c' > 0$. Thus, again by Assumption 6,

$$
\lim_{K \to \infty} \sum_{k=0}^{K} \alpha_k^2 \gamma_k^2 < \infty.
\tag{45}
$$

We conclude that

$$
\lim_{K \to \infty} \sum_{k=0}^{K} \alpha_k \gamma_k \|\nabla F(\theta_k)\|^2 < \infty.
\tag{46}
$$

Moreover, since the series $\sum_k \alpha_k \gamma_k$ diverges by Assumption 6, we have

$$
\lim_{k \to \infty} \inf \|\nabla F(\theta_k)\| = 0.
\tag{47}
$$

To prove that $\lim_{k \to \infty} \|\nabla F(\theta_k)\| = 0$, we consider the hypothesis H) $\lim_{k \to \infty} \sup \|\nabla F(\theta_k)\| \ge \rho$ for an arbitrary $\rho > 0$.

Assume (H) to be true. Then, we can always find an arbitrary subsequence $\left( \|\nabla F(\theta_{k_l})\| \right)_{l \in \mathbb{N}}$ of $\|\nabla F(\theta_k)\|$, such that $\|\nabla F(\theta_{k_l})\| \ge \rho - \varepsilon, \forall l$, for $\rho - \varepsilon > 0$ and $\varepsilon > 0$.

Then, by the $L$-smoothness property and applying the descent step of the algorithm,

$$
\begin{aligned}
\|\nabla F(\theta_{k_l+1})\| &\ge \|\nabla F(\theta_{k_l})\| - \|\nabla F(\theta_{k_l+1}) - \nabla F(\theta_{k_l})\| \\
&\ge \rho - \varepsilon - L\|\theta_{k_l+1} - \theta_{k_l}\| \\
&= \rho - \varepsilon - L\alpha_{k_l} \|g_{k_l}^{(2P)}\| \\
&\ge \rho - \varepsilon - L\sqrt{c'} \alpha_{k_l} \gamma_{k_l},
\end{aligned}
\tag{48}
$$

Since $k_l \to \infty$ as $l \to \infty$, we can always find a subsequence of $(k_{l_p})_{p \in \mathbb{N}}$ such that $k_{l_{p+1}} - k_{l_p} > 1$. As $\alpha_{k_l}\gamma_{k_l}$ is vanishing, we consider $(k_l)_{l \in \mathbb{N}}$ starting from $\alpha_{k_l}\gamma_{k_l} < \frac{\rho - \varepsilon}{L\sqrt{c'}}$. Thus,

$$
\begin{aligned}
&\sum_{k=0}^{\infty} \alpha_{k+1}\gamma_{k+1}\|\nabla F(\theta_{k+1})\|^2 \\
&\geq (\rho - \varepsilon)^2 \sum_{k=0}^{\infty} \alpha_{k+1}\gamma_{k+1} - 2(\rho - \varepsilon)L\sqrt{c'}\sum_{k=0}^{\infty}\alpha_{k+1}\gamma_{k+1}\alpha_k\gamma_k + L^2 c'\sum_{k=0}^{\infty}\alpha_{k+1}\gamma_{k+1}\alpha_k^2\gamma_k^2 \\
&\geq (\rho - \varepsilon)^2 \sum_{k=0}^{\infty} \alpha_{k+1}\gamma_{k+1} - 2(\rho - \varepsilon)L\sqrt{c'}\sum_{k=0}^{\infty}\alpha_k^2\gamma_k^2 + L^2 c'\sum_{k=0}^{\infty}\alpha_{k+1}\gamma_{k+1}\alpha_k^2\gamma_k^2 \\
&= +\infty,
\end{aligned}
\tag{49}
$$

as the first series diverges, and the second and the third converge by Assumption 6. This implies that the series $\sum_k \alpha_k\gamma_k\|\nabla F(\theta_k)\|^2$ diverges. This is a contradiction as this series converges almost surely by (46). Therefore, hypothesis (H) cannot be true and $\|\nabla F(\theta_k)\|$ converges to zero almost surely.

### D.3 Proof of Theorem 4: Convergence rate

Due to the $L$-smoothness, we have

$$
F(\theta_{k+1}) \leq F(\theta_k) - \alpha_k\langle\nabla F(\theta_k), g_k^{(2P)}\rangle + \frac{\alpha_k^2 L}{2}\|g_k^{(2P)}\|^2.
\tag{50}
$$

Taking the conditional expectation given $\mathcal{H}_k$,

$$
\begin{aligned}
F(\theta_{k+1}) &\leq F(\theta_k) - c_1'\alpha_k\gamma_k\langle\nabla F(\theta_k), \nabla F(\theta_k) + b_k'\rangle + \frac{c_2'L}{2}\alpha_k^2\gamma_k^2 \\
&= F(\theta_k) - c_1'\alpha_k\gamma_k\|\nabla F(\theta_k)\|^2 - c_1'\alpha_k\gamma_k\langle\nabla F(\theta_k), b_k'\rangle + \frac{c_2'L}{2}\alpha_k^2\gamma_k^2 \\
&\overset{(a)}{\leq} F(\theta_k) - c_1'\alpha_k\gamma_k\|\nabla F(\theta_k)\|^2 + \frac{c_1'\alpha_k\gamma_k}{2}\|\nabla F(\theta_k)\|^2 + \frac{c_1'\alpha_k\gamma_k}{2}\|b_k'\|^2 + \frac{c_2'L}{2}\alpha_k^2\gamma_k^2 \\
&= F(\theta_k) - \frac{c_1'\alpha_k\gamma_k}{2}\|\nabla F(\theta_k)\|^2 + \frac{c_1\alpha_k\gamma_k}{2}\|b_k'\|^2 + \frac{c_2'L}{2}\alpha_k^2\gamma_k^2
\end{aligned}
\tag{51}
$$

where $(a)$ is by $-\langle a, b\rangle \leq \frac{1}{2}\|a\|^2 + \frac{1}{2}\|b\|^2$

We again consider a large value $K > 0$ and take the telescoping sum of (51),

$$
\begin{aligned}
\mathbb{E}[F(\theta_{K+1})|\mathcal{H}_K] &\leq F(\theta_0) - \frac{c_1'}{2}\sum_k \alpha_k\gamma_k\|\nabla F(\theta_k)\|^2 + \frac{c_1'}{2}\sum_k \alpha_k\gamma_k\|b_k'\|^2 + \frac{c_2'L}{2}\sum_k \alpha_k^2\gamma_k^2 \\
0 \leq \mathbb{E}[\delta_{K+1}|\mathcal{H}_K] &\leq \delta_0 - \frac{c_1'}{2}\sum_k \alpha_k\gamma_k\|\nabla F(\theta_k)\|^2 + \frac{c_1'}{2}\sum_k \alpha_k\gamma_k\|b_k'\|^2 + \frac{c_2'L}{2}\sum_k \alpha_k^2\gamma_k^2
\end{aligned}
\tag{52}
$$

Hence,

$$
\sum_k \alpha_k\gamma_k\mathbb{E}[\|\nabla F(\theta_k)\|^2] \leq \frac{2}{c_1'}\delta_0 + c_3'\sum_k \alpha_k\gamma_k^3 + \frac{c_2'L}{c_1'}\sum_k \alpha_k^2\gamma_k^2
\tag{53}
$$

Let the step sizes have the form as in Example 3. We know that, $\forall K > 0$,

$$
\begin{aligned}
\sum_{k=0}^{K} \alpha_k \gamma_k^3 = \alpha_0 \gamma_0^3 + \sum_{k=1}^{K} \alpha_k \gamma_k^3 &\leq \alpha_0 \gamma_0^3 \left( 1 + \int_0^K (x+1)^{-v_1 - 3v_2} dx \right) \\
&= \alpha_0 \gamma_0^3 \left( 1 + \frac{1}{v_1 + 3v_2 - 1} - \frac{(K+1)^{-v_1 - 3v_2 + 1}}{v_1 + 3v_2 - 1} \right) \\
&\leq \alpha_0 \gamma_0^3 \left( 1 + \frac{1}{v_1 + 3v_2 - 1} \right) \\
&= \alpha_0 \gamma_0^3 \left( \frac{v_1 + 3v_2}{v_1 + 3v_2 - 1} \right).
\end{aligned}
\tag{54}
$$

Similarly,

$$
\sum_{k=0}^{K} \alpha_k^2 \gamma_k^2 \leq \alpha_0^2 \gamma_0^2 \left( \frac{2v_1 + 2v_2}{2v_1 + 2v_2 - 1} \right)
\tag{55}
$$

- Next, when $0 < v_1 + v_2 < 1$,

$$
\begin{aligned}
\sum_{k=0}^{K} \alpha_k \gamma_k &\geq \alpha_0 \gamma_0 \int_0^{K+1} (x+1)^{-v_1 - v_2} dx \\
&= \frac{\alpha_0 \gamma_0}{(1 - v_1 - v_2)} \left( (K+2)^{1 - v_1 - v_2} - 1 \right).
\end{aligned}
\tag{56}
$$

Thus, substituting in inequality (53), we get

$$
\begin{aligned}
\frac{\sum_k \alpha_k \gamma_k \mathbb{E}[\|\nabla F(\theta_k)\|^2]}{\sum_k \alpha_k \gamma_k} \leq & \frac{(1 - v_1 - v_2)}{(K+2)^{1 - v_1 - v_2} - 1} \\
&\times \left( \frac{2\delta_0}{c_1' \alpha_0 \gamma_0} + \frac{c_3'^2 \gamma_0^2 (v_1 + 3v_2)}{v_1 + 3v_2 - 1} + \frac{2 c_2' L \alpha_0 \gamma_0 (v_1 + v_2)}{c_1' (2v_1 + 2v_2 - 1)} \right).
\end{aligned}
\tag{57}
$$

- Otherwise, when $v_1 + v_2 = 1$,

$$
\begin{aligned}
\sum_{k=0}^{K} \alpha_k \gamma_k &\geq \alpha_0 \gamma_0 \int_0^{K+1} \frac{1}{x+1} dx \\
&= \alpha_0 \gamma_0 \ln(K+2).
\end{aligned}
\tag{58}
$$

Then, we obtain

$$
\begin{aligned}
\frac{\sum_k \alpha_k \gamma_k \mathbb{E}[\|\nabla F(\theta_k)\|^2]}{\sum_k \alpha_k \gamma_k} \leq & \frac{1}{\ln(K+2)} \\
&\times \left( \frac{2\delta_0}{c_1' \alpha_0 \gamma_0} + \frac{c_3'^2 \gamma_0^2 (v_1 + 3v_2)}{v_1 + 3v_2 - 1} + \frac{2 c_2' L \alpha_0 \gamma_0 (v_1 + v_2)}{c_1' (2v_1 + 2v_2 - 1)} \right).
\end{aligned}
\tag{59}
$$

# E  EXPERIMENTAL RESULTS DETAILS

## E.1  AUTOENCODER

The lossy autoencoder that we train to compress the images of the MNIST dataset has an encoder-decoder architecture. The encoder part takes an input of size 784 and gradually reduces its dimensionality to 10. This is accomplished by passing the input tensor through three linear layers whose dimensions move from 784 to 512, to 128, to 10, with the first two layers followed by an ELU activation function.

The decoder part takes the encoded representation and reconstructs the original input tensor by performing the reverse operation of the encoder. Specifically, the encoded tensor is passed through three linear layers whose sizes move from 10 to 128, to 512 to 784, with the first two layers followed by an ELU activation function and, finally, a sigmoid activation function in the last layer to ensure that the output pixel intensities are between 0 and 1.

Overall, the autoencoder architecture consists of 6 linear layers and 4 activation functions (3 ELU and 1 sigmoid).

We train the model on all the images of the MNIST dataset over 10 epochs using the mean squared error loss and Adam optimizer.

### E.2 PARAMETERS CHOICE

$\Phi_k$ is generated according to Example 1. All channels are generated using the normal distribution with autocovariance $K_{hh} = \frac{1}{2}$. The noise is Gaussian with 0 mean and variance $\sigma_n^2 = \frac{1}{4}$.

1. For the logistic regression model, we consider the following step sizes/ learning rates for every algorithm:
   - 1P-ZOFL: $\alpha_k = (1 + k)^{-0.51}$ and $\gamma_k = 3(1 + k)^{-0.18}$
   - 2P-ZOFL: $\alpha_k = 3(1 + k)^{-0.26}$ and $\gamma_k = 6(1 + k)^{-0.26}$
   - FedAvg: $\eta = 0.15$

   With the non-IID data distribution, we only change the following:
   - 2P-ZOFL: $\alpha_k = 3.5(1 + k)^{-0.26}$ and $\gamma_k = 6.5(1 + k)^{-0.26}$

2. For the training model, we consider the following:
   - 1P-ZOFL: $\alpha_k = 0.1(1 + k)^{-0.51}$ and $\gamma_k = 0.3(1 + k)^{-0.18}$
   - 2P-ZOFL: $\alpha_k = 0.4(1 + k)^{-0.26}$ and $\gamma_k = 0.7(1 + k)^{-0.26}$
   - FedAvg: $\eta = 0.01$

### E.3 QUANTITATIVE COMPARISON

Table 1: Upload communication efficiency of ZOFL vs FedAvg (McMahan et al., 2017) vs FedZO (Fang et al., 2022) till convergence and per iteration.

| ALGORITHM | TOTAL SYMBOLS UNTIL CONVERGENCE FOR 1 DEVICE | TOTAL SYMBOLS UNTIL CONVERGENCE FOR 100 DEVICES | NUMBER OF SYMBOLS PER ITERATION FOR 1 DEVICE | NUMBER OF SYMBOLS PER ITERATION FOR 100 DEVICES |
|---|---|---|---|---|
| ZOFL | $4,000$ | $400,000$ | $2$ | $200$ |
| FEDAVG | $59,280,600$ | $5,928,060,000$ | $59,280,600/300 = 197,602$ | $19,760,200$ |
| FEDZO | $59,280,600/50 = 1,185,612$ | $1,185,612 \times 10 = 11,856,120$ | $1,185,612/300 = 3,952$ | $39,520$ |

We include this quantitative study to compare with other communication-efficient strategies, like local SGD (multiple local gradient descent steps before upload) and partial device participation at every iteration. For this study, we compare with FedZO (Fang et al., 2022), which incorporates both strategies and communicates over wireless channels.

Casting the added complexity aside for now (for the extra channel processing, piloting, and training time needed) for FedZO and assuming that FedZO with $H = 50$ (local updates) and $M = 10$ (10% of the users) performs as well as FedAvg presented in our experiments (which is quite generous), the number of scalar values to collectively upload is still very high ($59280600/500 = 118561$) as compared to that of our algorithms. If we add the additional time needed for local training (50 times forward and backward passes through the local models) and the back-and-forth exchange for instantaneous channel coefficients and maximum squared norm of local models between the server

and clients in section IV.B of Fang et al. (2022), then the communication efficiency of our algorithm will be clearly evident as compared to the literature. We present the number of symbols to upload in Table 1 for an easier read of comparison.

### E.4 Performance of 1P-ZOFL vs SNR

An important remark to convey is that our algorithm is independent of the amount of noise present in the wireless environment. High SNR is needed when we must decode the information in the received signal. In our case, nothing is decoded; there is no channel estimation or gradient calculation from the received signal. Rather, the received signal is fed directly into the learning (the channel is part of the learning). The amount of noise present in the system does not affect our algorithms' convergence: Examining (15) and (17), we see that the noise variance $\sigma_n^2$ might only increase the norms of the estimate and bias, but if we refer to (19), we'll find that both terms are multiplied by step sizes, and we can counter the noise effect by decreasing the step sizes' constant parts (i.e, in the terms $\frac{c_1 \alpha_k \gamma_k}{2}\|b_k\|^2$ and $\frac{\alpha_k^2 L}{2}\|g_k^{(1P)}\|^2$). We have provided numerical evidence in Figure 4 to assess this explanation. In the plots, we decreased $\alpha_0$ and $\gamma_0$ when we increased $\sigma_n^2$. When taking the noise variance equal to 1, i.e. same variance as the channel's, we get the same performance. When taking it 2.25, which refers to poor communication quality, the algorithm was able to keep up when the step sizes became small enough, proving a difference in the rate affected in its constant part. Finally, they all converged to the same accuracy, meaning there was no gap in the convergence due to the noise.

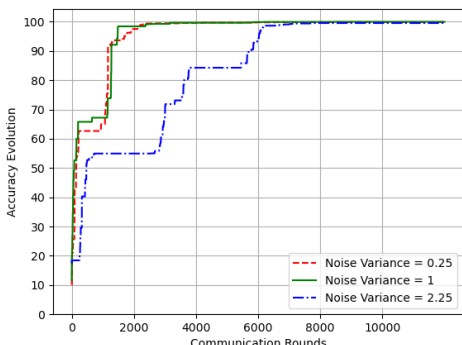

Figure 4: 1P-ZOFL with logistic regression and different noise variance $\sigma_n^2$.

## F    Non-Symmetrical Channels

Assuming non-symmetrical channels with $\mathbb{E}[h_{i,k}] = \mu_h$ and $\sigma_h^2 = \mathbb{E}[h_{i,k}^2] - \mu_h^2, \forall i, \forall k$, the one-point gradient estimate becomes

$$g_k^{(1P)} = \Phi_k \sum_{i=1}^{N} \left[ h_{i,k+1} \tilde{f}_i (\theta_k + \gamma_k \Phi_k, S_{i,k+1}) + n_{i,k+1} \right]. \tag{60}$$

And 1P-ZOFL is updated to Algorithm 3.

The two-point estimator also changes to

$$g_k^{(2P)} = \Phi_k \sum_{i=1}^{N} h_{i,k+1} \left[ \tilde{f}_i \left( \theta_k + \gamma_k \Phi_k, S_{i,k+1} \right) - \tilde{f}_i \left( \theta_k - \gamma_k \Phi_k, S_{i,k+1} \right) \right]. \tag{61}$$

And 2P-ZOFL is updated to Algorithm 4.

---

**Algorithm 3** The 1P-ZOFL algorithm with non-symmetrical channels

---

**Input:** Initial model $\theta_0 \in \mathbb{R}^d$, the initial step-sizes $\alpha_0$ and $\gamma_0$, and the channels' standard deviation $\sigma_h$

1: **for** $k = 0, 2, 4, ...$ **do**
2:     The server broadcasts $\theta_k + \gamma_k \Phi_k$
3:     The server receives $\sum_{i=1}^{N} h_{i,k+1} \tilde{f}_i(\theta_k + \gamma_k \Phi_k, S_{i,k+1}) + n_{i,k+1}$
4:     The server multiplies the received scalar sum by $\Phi_k$ to assemble $g_k^{(1P)}$ in (60)
5:     The server updates $\theta_{k+1} = \theta_k - \alpha_k g_k^{(1P)}$
6: **end for**

---

**Algorithm 4** The 2P-ZOFL algorithm with non-symmetrical channels

---

**Input:** Initial model $\theta_0 \in \mathbb{R}^d$, the initial step-sizes $\alpha_0$ and $\gamma_0$, and the channels' standard deviation $\sigma_h$

1: **for** $k = 0, 2, 4, ...$ **do**
2:     The server broadcasts $\theta_k + \gamma_k \Phi_k$ and $\theta_k - \gamma_k \Phi_k$ under the same stochastic wireless environment
3:     The server receives
$$\sum_{i=1}^{N} h_{i,k+1} \left[ \tilde{f}_i(\theta_k + \gamma_k \Phi_k, S_{i,k+1}) - \tilde{f}_i(\theta_k - \gamma_k \Phi_k, S_{i,k+1}) \right]$$
4:     The server multiplies the received scalar sum by $\Phi_k$ to assemble $g_k^{(2P)}$ in (61)
5:     The server updates $\theta_{k+1} = \theta_k - \alpha_k g_k^{(2P)}$
6: **end for**

---

We then analyze the properties of our modified gradient estimates:

$$
\begin{aligned}
&\mathbb{E}[g_k^{(1P)}|\mathcal{H}_k] \\
=&\mathbb{E}\left[ \Phi_k \sum_{i=1}^{N} \left( h_{i,k+1} \tilde{f}_i(\theta_k + \gamma_k \Phi_k, S_{i,k+1}) + n_{i,k+1} \right) \Big| \mathcal{H}_k \right] \\
=&\mathbb{E}\left[ \Phi_k \sum_{i=1}^{N} \mu_h \tilde{F}_i(\theta_k + \gamma_k \Phi_k) \Big| \mathcal{H}_k \right] \\
=&\mathbb{E}\left[ \Phi_k \mu_h \sum_{i=1}^{N} \left( \tilde{F}_i(\theta_k) + \gamma_k \Phi_k^T \nabla \tilde{F}_i(\theta_k) + \gamma_k^2 \Phi_k^T \nabla^2 \tilde{F}_i(\breve{\theta}_k) \Phi_k \right) \Big| \mathcal{H}_k \right] \\
=&\mu_h \gamma_k \sum_{i=1}^{N} \mathbb{E}\left[ \Phi_k \Phi_k^T \nabla \tilde{F}_i(\theta_k) + \gamma_k \Phi_k \Phi_k^T \nabla^2 \tilde{F}_i(\breve{\theta}_k) \Phi_k \Big| \mathcal{H}_k \right] \\
=&c_1 \gamma_k (\nabla F(\theta_k) + b_k),
\end{aligned}
\tag{62}
$$

with $c_1 = \frac{\mu_h \alpha_2}{\sigma_h^2}$ and $b_k = \frac{\gamma_k}{\alpha_2} \mathbb{E}\left[ \Phi_k \Phi_k^T \nabla^2 F_i(\breve{\theta}_k) \Phi_k \Big| \mathcal{H}_k \right]$.

Then, $\|b_k\| \leq \frac{\gamma_k}{\alpha_2} \alpha_3^3 \alpha_1$, now $c_3 = \frac{\alpha_3^3 \alpha_1}{\alpha_2}$ and $\|b_k\| \leq c_3 \gamma_k$.

$$\mathbb{E}[\|g_k^{(1P)}\|^2|\mathcal{H}_k]$$

$$=\mathbb{E}\left[\left\|\Phi_k\sum_{i=1}^{N}h_{i,k+1}\left[\tilde{f}_i\Big(\theta_k+\gamma_k\Phi_k,S_{i,k+1}\Big)-\tilde{f}_i\Big(\theta_k-\gamma_k\Phi_k,S_{i,k+1}\Big)\right]\right\|^2\Big|\mathcal{H}_k\right]$$

$$=\mathbb{E}\left[\|\Phi_k\|^2\Big(\sum_{i=1}^{N}h_{i,k+1}\tilde{f}_i\Big(\theta_k+\gamma_k\Phi_k,S_{i,k+1}\Big)+n_{i,k+1}\Big)^2\Big|\mathcal{H}_k\right]$$

$$\leq\alpha_3^2 N\sum_{i=1}^{N}\mathbb{E}\left[\Big(h_{i,k+1}\tilde{f}_i\Big(\theta_k+\gamma_k\Phi_k,S_{i,k+1}\Big)+n_{i,k+1}\Big)^2\Big|\mathcal{H}_k\right]$$

$$=\alpha_3^2 N\sum_{i=1}^{N}\mathbb{E}\left[h_{i,k+1}^2\tilde{f}_i^2\Big(\theta_k+\gamma_k\Phi_k,S_{i,k+1}\Big)+n_{i,k+1}^2\Big|\mathcal{H}_k\right]$$

$$\leq\alpha_3^2 N\sum_{i=1}^{N}\mathbb{E}\left[\frac{h_{i,k+1}^2}{\sigma_h^4}\Big(\|f_i(0,S_{i,k+1})\|+L_{S_{i,k+1}}\|\theta_k+\gamma_k\Phi_k\|\Big)^2\Big|\mathcal{H}_k\right]+N^2\alpha_3^2\sigma_n^2$$

$$\leq\alpha_3^2 N\sum_{i=1}^{N}\mathbb{E}\left[\frac{h_{i,k+1}^2}{\sigma_h^4}\Big(\|f_i(0,S_{i,k+1})\|+L_{S_{i,k+1}}\|\theta_k\|+L_{S_{i,k+1}}\gamma_k\|\Phi_k\|\Big)^2\Big|\mathcal{H}_k\right]+N^2\alpha_3^2\sigma_n^2$$

$$\leq 3\alpha_3^2 N\sum_{i=1}^{N}\mathbb{E}\left[\frac{h_{i,k+1}^2}{\sigma_h^4}\Big(\|f_i(0,S_{i,k+1})\|^2+L_{S_{i,k+1}}^2\|\theta_k\|^2+\alpha_3^2 L_{S_{i,k+1}}^2\gamma_k^2\Big)\Big|\mathcal{H}_k\right]+N^2\alpha_3^2\sigma_n^2$$

$$=\frac{3N^2\alpha_3^2(\sigma_h^2+\mu_h^2)}{\sigma_h^2}\Big(\mu_S+L_S\|\theta_k\|^2+\alpha_3^2 L_S\gamma_k^2\Big)+N^2\alpha_3^2\sigma_n^2$$

$$:=c_2,$$

$$(63)$$

For the two-point estimate,

$$\mathbb{E}[g_k^{(2P)}|\mathcal{H}_k]$$

$$=\mathbb{E}\left[\Phi_k\sum_{i=1}^{N}h_{i,k+1}\left[\tilde{f}_i\Big(\theta_k+\gamma_k\Phi_k,S_{i,k+1}\Big)-\tilde{f}_i\Big(\theta_k-\gamma_k\Phi_k,S_{i,k+1}\Big)\right]\Big|\mathcal{H}_k\right]$$

$$=\mathbb{E}\left[\Phi_k\sum_{i=1}^{N}\mu_h\left[\tilde{F}_i\Big(\theta_k+\gamma_k\Phi_k\Big)-\tilde{F}_i\Big(\theta_k-\gamma_k\Phi_k\Big)\right]\Big|\mathcal{H}_k\right]$$

$$=\mathbb{E}\left[\Phi_k\sum_{i=1}^{N}\mu_h\Big[\tilde{F}_i(\theta_k)+\gamma_k\Phi_k^T\nabla\tilde{F}_i(\theta_k)+\gamma_k^2\Phi_k^T\nabla^2\tilde{F}_i(\acute{\theta}_k)\Phi_k\right.$$

$$\left.-\Big(\tilde{F}_i(\theta_k)-\gamma_k\Phi_k^T\nabla\tilde{F}_i(\theta_k)+\gamma_k^2\Phi_k^T\nabla^2\tilde{F}_i(\grave{\theta}_k)\Phi_k\Big)\Big]\Big|\mathcal{H}_k\right]$$

$$(64)$$

$$=2\gamma_k\mu_h\sum_{i=1}^{N}\mathbb{E}\left[\Phi_k\Phi_k^T\nabla\tilde{F}_i(\theta_k)+\frac{\gamma_k}{2}\Phi_k^T(\nabla^2\tilde{F}_i(\acute{\theta}_k)-\nabla^2\tilde{F}_i(\grave{\theta}_k))\Phi_k\Big|\mathcal{H}_k\right]$$

$$=c_1'\gamma_k(\nabla F(\theta_k)+b_k')$$

with $c_1'=2\frac{\mu_h\alpha_2}{\sigma_h^2}$ and $b_k'=\frac{\gamma_k}{2\alpha_2}\mathbb{E}\left[\Phi_k\Phi_k^T(\nabla^2\tilde{F}_i(\acute{\theta}_k)-\nabla^2\tilde{F}_i(\grave{\theta}_k))\Phi_k\Big|\mathcal{H}_k\right]$.

Then, $\|b_k'\|\leq\frac{\gamma_k}{2\alpha_2}\alpha_3^3\alpha_1$, now $c_3'=\frac{\alpha_3^3\alpha_1}{2\alpha_2}$ and $\|b_k'\|\leq c_3'\gamma_k$.

$$
\mathbb{E}[\|g_k^{(2P)}\|^2|\mathcal{H}_k]
$$

$$
=\mathbb{E}\left[\left\|\Phi_k\sum_{i=1}^N h_{i,k+1}\left[\tilde{f}_i\Big(\theta_k+\gamma_k\Phi_k,S_{i,k+1}\Big)-\tilde{f}_i\Big(\theta_k-\gamma_k\Phi_k,S_{i,k+1}\Big)\right]\right\|^2\Big|\mathcal{H}_k\right]
$$

$$
=\mathbb{E}\left[\|\Phi_k\|^2\Big(\sum_{i=1}^N h_{i,k+1}\left[\tilde{f}_i\Big(\theta_k+\gamma_k\Phi_k,S_{i,k+1}\Big)-\tilde{f}_i\Big(\theta_k-\gamma_k\Phi_k,S_{i,k+1}\Big)\right]\Big)^2\Big|\mathcal{H}_k\right]
$$

$$
\leq\alpha_3^2\mathbb{E}\left[\Big(\sum_{i=1}^N h_{i,k+1}\left[\tilde{f}_i\Big(\theta_k+\gamma_k\Phi_k,S_{i,k+1}\Big)-\tilde{f}_i\Big(\theta_k-\gamma_k\Phi_k,S_{i,k+1}\Big)\right]\Big)^2\Big|\mathcal{H}_k\right]
$$

$$
\leq\alpha_3^2\mathbb{E}\left[\Big(\sum_{i=1}^N \frac{h_{i,k+1}}{\sigma_h^2}L_{S_{i,k+1}}\Big\|2\gamma_k\Phi_k\Big\|\Big)^2\Big|\mathcal{H}_k\right]
$$

$$
\leq\frac{4\gamma_k^2\alpha_3^4 N}{\sigma_h^4}\sum_{i=1}^N\mathbb{E}\left[h_{i,k+1}^2 L_{S_{i,k+1}}^2\Big|\mathcal{H}_k\right]
$$

$$
=4\gamma_k^2\frac{\alpha_3^4(\sigma_h^2+\mu_h^2)L_S N^2}{\sigma_h^4}
$$

$$
:=c_2'\gamma_k^2
$$

(65)

where $L_S=\mathbb{E}[L_{S_{i,k+1}}|\mathcal{H}_k]$ and $c_2'=4\frac{\alpha_3^4(\sigma_h^2+\mu_h^2)L_S N^2}{\sigma_h^4}$.

The rest of the analysis resumes as previously.

