# OpenReview forum: "Rendering Wireless Environments Useful for Gradient Estimators: A Zero-Order Stochastic Federated Learning Method"
_ICLR.cc/2024/Conference — Submitted to ICLR 2024_

### Official Review · Reviewer_DD4A · 2023-11-02

**Soundness:** 2 fair
**Presentation:** 3 good
**Contribution:** 3 good
**Rating:** 5
**Confidence:** 4

**Summary:**

The authors first propose a zero-order method with two types of gradient estimators to solve the federated learning problems in the wireless communication environment. The proposed framework ZOFL doesn’t require the knowledge of channel state. Moreover, the authors prove the almost surely convergence and give the convergence rate. Finally, the authors test the performance of ZOFL with experimental results.

**Strengths:**

The proposed algorithm ZOFL is novel and interesting in my point of view. (1) Despite using the zeroth-order method in optimization problems is not a new thing, the proposed algorithm includes the channel state as a part of learning and doesn’t need to analyze the channel, which has not been seen in the literature. (2) Theoretically, the proposed paper proves the almost surely convergence instead of convergence in probability by utilizing Doob’s martingale inequality.

**Weaknesses:**

Despite that the paper proposes an attractive novelty, I suspect some technical proofs have some small problems, which I list in the questions part. It is possible that I’m wrong. Thus, if the authors can explain these questions, I will change my score.

**Questions:**

(1) In the last line of Eq. (15), How E[h_{i,k+1}^2 h_{i,k}^2] be bounded by \sigma_h^4? Shouldn’t any two of them be related? Similarly, how E[h_{i,k+1}^2 \sum_{j<l}h_{j,k}h_{l,k}] be bounded by K_{hh}^2 since j and k cannot be equal to i at the same time?

(2) In the appendix C1 page 18, whey g_k and g_k’ are independent if k\neq k’? since g_k includes h_k and h_{k+1}. If k’=k+1, shouldn’t they be related? This problem also influences the derivation of the following proofs which use Doob’s martingale inequality.

(3) In the Eq (27) on page 20, how can you guarantee that \rho-\eps-L\sqrt{c}\alpha_{k_l} is positive? If not, the inequality cannot be squared and used in Eq. (28) I think.

Besides the above questions, this paper also has some typos, which I list here.

(1)	In equation (11) equality (c), where does the ‘2N’ come from in the second term?
(2)	In equation (12), a vector \Phi_k is missing before the Hessian matrix.
(3)	In equation (15) inequality (a) where does the ‘N’ come from? Giving the assumption 2 ||\Phi_k||^2 should be directly bounded by \alpha_3^2. I don’t think Cauchy Schwarz works here.
(4)	1/\nu^2 is missing in inequality (a) on page 19 in Appendix C1.
(5)	In Eq. (30) the conditional expectation notation is missing.

---

> ### Author Response · Authors · 2023-11-22
>
> We thank the reviewer for the constructive review and we truly appreciate taking the time to look into the details of the Appendices. We also highly appreciate your acknowledgement of our paper’s contribution and we hope that this rebuttal clarifies and answers the raised questions.
>
> To answer your concerns:
>
> 1-	We apologize for the confusion in the writing of the proof. In Eq. (15), $\mathbb{E}[h_{i,k+1}^2 h_{i,k}^2]$ is not bounded by $\sigma_h^4$, it is equal to both terms $\sigma_{h}^4+2K_{hh}^2$ (similarly to a multivariate normal distribution) and $\mathbb{E}[h_{i,k+1}^2 \sum_{j\neq i} h_{j,k}^2]$ is equal to $(N-1)\sigma_{h}^4$ as $ h_{i,k+1}$ and $h_{j,k}$ are independent. The term $\mathbb{E}[h_{i,k+1}^2 \sum_{j<l } h_{j,k}h_{l,k}]$ is equal to zero as one of $ h_{j,k}$ and $ h_{l,k}$ will always be independent of the other terms in the product and has a mean equal to zero.
>
> 2-	We apologize for the misleading presentation of the martingale definition. In fact, $g_k$ and $g_{k’}$ don’t need to be independent for $k \neq k’$, we only need the stochastic errors at iterations $k$ and $k’$ to be uncorrelated, not independent and this is satisfied as for every $k’<k$, we can always take the law of total expectation:
>  $
> \mathbb{E}\big[\mathbb{E}[e_k^T e_{k’}|\mathcal{H}\_k]\big] = \mathbb{E}\big[\mathbb{E}[e_k^T |\mathcal{H}\_k] e_{k’} \big] = 0.
> $
>
> To clarify, $\\{ \sum_{k=K}^{K'} \alpha_k e_k \\}\_{K'\geq K}$    is a martingale as for all    $K’\geq K$, $X_{K’}=\sum_{k=K}^{K'} \alpha_k e_k$ satisfies the following two conditions:
>
> (a)	$\mathbb{E}[ X_{K’+1}|X_{K’}]=\mathbb{E}[\alpha_{K’+1} e_{K’+1}+ \sum_{k=K}^{K'} \alpha_k e_k |\sum_{k=K}^{K'} \alpha_k e_k] = 0+ \sum_{k=K}^{K'} \alpha_k e_k = X_{K'}$
>
> (b)	$\mathbb{E}[\|\|X_{K’}\|\|^2]<\infty$
>
> The proof of (b) follows from the exact same inequalities we use in Appendix C.1 page 19 without the factor $ \frac{1}{\nu^2} $ and it’s due to the uncorrelation of the stochastic errors at different iterations and $\sum_k \alpha_k^2 < \infty$.
>
> 3-	We stated right before eq. (28) “As $\alpha\_{k\_l}$ is vanishing,
> we consider  $(k\_l)\_{l\in\mathbb{N}}$ starting from  $\alpha\_{k\_l} < \frac{\rho - \varepsilon}{L\sqrt{c}}$.”
>
> So, we only consider the subsequence $(k\_l)\_{l\in\mathbb{N}}$ starting from the point
> $\rho-\epsilon-L\sqrt{c}\alpha_{k_l}>0$.
>
> We thank you so much for pointing out the typos. Just to clarify,
>
> In eq. (11), equality (c) should be an inequality with $\Big(\sum_{j=1}^{N}\frac{h_{j,k}}{\sigma_h^2}+n_{j,k}\Big)^2 \leq N \sum_{j=1}^{N}\Big(\frac{h_{j,k}^2}{\sigma_h^4}+n_{j,k}^2\Big)$ and we should remove the scalar $2$ from $2N$. This inequality comes from the Cauchy-Schwarz for any real $a_i$:
>
> $\Big(\sum\_{j=1}^{N}1\cdot a\_i\Big)^2 \leq \Big( \sum\_{j=1}^{N} a\_i^2 \Big) \cdot \Big( \sum\_{j=1}^{N} 1^2 \Big)  = N \Big( \sum\_{j=1}^{N} a\_i^2 \Big) $.
>
> In eq.(12), you are correct, thank you for pointing it out.
>
> In eq. (15) inequality (a), N comes from Cauchy-Schwarz similar to what’s explained above.
>
> On page 19, you are correct $ \frac{1}{\nu^2} $ is missing from inequality (a), thank you so much for pointing it out.
>
> In Eq. (30) the conditional expectation notation is missing, you are correct. Thank you again for pointing it out.

---

### Official Review · Reviewer_b8PD · 2023-11-03

**Soundness:** 3 good
**Presentation:** 2 fair
**Contribution:** 2 fair
**Rating:** 3
**Confidence:** 4

**Summary:**

The authors have considered the federated learning problem over wireless channels. The proposed zero-order method optimizes this process over wireless channels, integrating channel characteristics directly into the algorithm. The authors have provided convergence analysis and experiments for the proposed algorithms.

**Strengths:**

- The paper is easy to read.

- The authors have provided theoretical analysis and experiments for the proposed techniques.

**Weaknesses:**

- The motivation of the work is not clear.
- The authors have mentioned a setting in which server and clients are communicating via wireless challenges, but the unique challenges due to this setting are not clearly mentioned.
- Why a zeroth order method would address the challenges due to wireless communication setting is not clear.
- The authors have motivated the use of single point estimates of gradients via mentioning that the ''settings are continuously changing over time", but it is not cleat what settings are referred to here?
- Also, in FL setting, the stochastic gradients a usually easily available, so what is the main motivation behind going for zeroth order optimization?
- It seems like the authors are trying to look at the physical layer aspect of federated learning? But does it require to change the FL algorithm ?
- The authors have mentioned about sharing the gradients with the server, but in most of the FL techniques, the core idea is to just share the models with the server, what is the motivation to share the gradients?
- Since the authors are considering to study the impact of wireless impact on FL, it would require to consider the wireless communication settings. In the wireless channel model Hg + n, g is usually the encoded bits which we transmit. But here the authors have used directly g, should it be something like b(g) instead of g directly?

- It is unclear why considering the effect of wireless channels directly in the algorithm updates would have additional benefits. On the other hand, it would add another challenge of getting to know the channel estimating for each device at each instant of transition.

- How are the convergence bounds related to other results in the literature such as with FedAvg, FedProx et?
- The analysis looks like follows directly from the existing analysis of FL techniques in the literature, what are the additional challenges?

**Questions:**

Please refer to weaknesses.

---

> ### Author Response · Authors · 2023-11-22
> **Comment (1/3)**
>
> We thank the reviewer for taking the time to read the paper and writing a review. Before replying to the comments and questions raised by the reviewer, we will first clarify the contribution of the paper as we believe it has not been well captured by the reviewer.
>
> In standard FL over wireless networks, the devices exchange their models or gradients (having usually high dimension) over the wireless channels. For that, the devices encode the long vector representing the model and transmit it to the server. This server has first to decode the information and then does the aggregation of the received models and then exchange the results with all devices. This encoding/decoding implies that the wireless channel coefficients are correctly estimated at each time slot (in practice it is 1ms). This consumes nonnegligible time, computation and power resources. Furthermore, the high dimension of the model requires too much communication overhead over wireless channels. The main motivation of this work is to develop an efficient modified FL method by: i) exchanging scalars instead of high dimension vectors by employing an efficient Zero order method, and ii) including the channel perturbations in the FL learning algorithm (avoiding thus channel estimation).  While there is clear interest of doing that, the difficulty of proving the convergence of the learning algorithm (that incorporates the channel perturbation) is not straightforward at all. We deal with a nonconvex setting, and zero-order method (instead of gradient in standard FL method). By using zero-order method, only a biased estimate of the gradient can be obtained (which complicates the convergence of the FL algorithm that requires exact/unbiased estimate of the gradient). In our case, in addition, the gradient biased estimate has an unbounded variance (while standard Zero order methods assume bounded variance). We proved mathematically that our proposed zero-order method converges almost surely to the correct model. Simulations results show also that the total communication overhead in our case is much less than the standard FL method. This is the main contribution of the paper, and, we invite the reviewer to check this contribution again (proofs and algorithm).  The aforementioned points, which are important to make a fair assessment of the contribution made in this paper, will be made clearer in the final paper.
>
> 1-  We proposed a new algorithm for federated learning over wireless networks by i) exchanging scalars (using a zero-order method), and ii)  integrating the channel in the learning itself (allowing hence to avoid estimating the channel at each time which will  save a lot of resources). Please see our explanations above for more details.
>
> 2- The challenges are already mentioned in the “communication bottleneck” and “channel impact” paragraphs. Federated learning over wireless channels constitutes a real issue in terms of uplink congestion. In addition to the uploaded optimization variable/gradient, the piloting (exchanges) needed to know the instantaneous channel state and decode properly, consumes a lot of these uplink resources. This work tries to alleviate these issues by providing an alternative to uploading the full gradient and to trying to guess the channel.
>
> 3- Alongside the utility of zero-order methods in general (when explicit gradient computation may be impractical, expensive, or impossible), the main idea of using a zeroth-order method in this paper is that we found a way to construct a gradient estimate by exchanging a scalar only over the wireless medium and to include the channel in the learning itself. We have overcome the need to decode, analyze, and remove the impact of the channel, which saves a lot of communication resources in addition to computation resources. This has never been done before.
>
> 4-	We meant the stochastic settings changing. As the loss function $ f(\theta, S)$ is subject to a perturbation $S$, to construct a two-point gradient estimate, the stochastic conditions, i.e., the stochastic variable $S$, have to remain unchanged during both queries of the loss function, i.e., during the calculation of both $ f(\theta + \gamma\Phi, S) $ and $ f(\theta - \gamma\Phi, S)$. This is not always the case in practical scenarios, especially in wireless networks when the stochastic environment continuously changes at a fast rate.

---

> > ### Author Response · Authors · 2023-11-22
> > **Comment (2/3)**
> >
> > 5-	The main motivation of Zero-order method is to counter the communication bottleneck. Instead of exchanging the model or the gradient of the loss function between the devices and the server, which is usually a long vector, one scalar (or two scalars) are only exchanged at each iteration between the device and the server, resulting thus in huge saving of communication resources. In addition, our method does not require estimating the channel at each time (resulting thus in an additional saving of resources), since the channel coefficient is included as a perturbation in the learning itself and our convergence results is robust to this perturbation. Other advantages of zero-order methods include black-box objective functions and applications where the gradient is impossible to compute, i.e., no explicit form of the objective function exists, like in tuning (and this happens in various contexts in practice).
> >
> > 6-	Yes, this integration of the channel into the learning algorithm changes the “structure” of the exchange. Our algorithm is carried out using a baseband signal exchange and steps including the coding and decoding of digital transmission are no longer needed. This is an alternative to digital transmission and we have introduced a new “structure” of exchange and proved mathematically its validity. This new structure now includes two scalar (symbol) uploads that are used to construct the estimate instead of sending the whole vector, and are aggregated directly without any coding/decoding or any other processing.
> >
> > 7-	Unless the model is divided between devices or some sort of processing is performed, the outcome of the learning process in federated learning isn’t affected by whether gradients or the entire model are uploaded. What’s the difference between updating the local model using the local gradient then aggregating the local models into the global model vs. updating the global model directly using the aggregation of the local gradients?? Moreover, the model and the gradient have the same dimensions, and that’s the issue we try to resolve in this paper: overcoming the communication bottleneck caused by this dimensionality.
> >
> > 8-	In this paper, the communication is analog and not digital. Hence, there’s no need to encode g into bits, i.e., b(g). Thus, the uploaded scalar (symbol) is uploaded directly, via analog communications. Recall that the main goal is to avoid extra processing (encoding, decoding, channel estimation) by incorporating the channel impact as a perturbation in the learning itself. This makes our algorithm robust to such perturbations and there is no need at all to encode into bits.
> >
> > It is worth mentioning that there is an increasing interest in the research community in avoiding digital encoding into bits when using ML over wireless (e.g. over the air computation [a], [b], [c], [d], ... )  to save resources as wireless systems cannot deal with the huge amount of data in this case.
> >
> > References:
> >
> > [a] G. Zhu, J. Xu, K. Huang, and S. Cui, "Over-the-Air Computing for Wireless Data Aggregation in Massive IoT," in IEEE Wireless Communications, vol. 28, no. 4, pp. 57-65, August 2021, doi: 10.1109/MWC.011.2000467.
> >
> > [b] G. Zhu, Y. Wang and K. Huang, "Broadband Analog Aggregation for Low-Latency Federated Edge Learning," in IEEE Transactions on Wireless Communications, vol. 19, no. 1, pp. 491-506, Jan. 2020, doi: 10.1109/TWC.2019.2946245.
> >
> > [c] W. Fang, Z. Yu, Y. Jiang, Y. Shi, C. N. Jones and Y. Zhou, "Communication-Efficient Stochastic Zeroth-Order Optimization for Federated Learning," in IEEE Transactions on Signal Processing, vol. 70, pp. 5058-5073, 2022, doi: 10.1109/TSP.2022.3214122.
> >
> > [d] K. Yang, T. Jiang, Y. Shi and Z. Ding, "Federated Learning via Over-the-Air Computation," in IEEE Transactions on Wireless Communications, vol. 19, no. 3, pp. 2022-2035, March 2020, doi: 10.1109/TWC.2019.2961673.

---

> > > ### Author Response · Authors · 2023-11-22
> > > **Comment (3/3)**
> > >
> > > 9-	We feel sorry that this concept was unclear. Throughout the paper, we’ve explained that there’s no need to estimate the channel and we’ve proved this mathematically. The convergence and the convergence rate of the algorithms do not require any knowledge of instantaneous channel information. We only use (statistical) information related to the mean and autocorrelation factor of the channel, which is based on the assumptions of the statistical channel distribution. No instantaneous estimation is needed whatsoever. We invite the reviewer to check the proofs in the appendix for elaboration. For information, the instantaneous channel estimation is rather needed in standard FL method when it is used over wireless systems (not in our method), since in that case, to decode the information (model or gradient) sent by the devices to the server, the server has to estimate the channel and to remove its impact at each time to perform the decoding.
> > >
> > > 10-	In general, stochastic gradient descent (SGD) with non-convex objective function and first-order information achieves a rate of $O(\frac{1}{\sqrt{K}})$ with vanishing step sizes, which is the same rate achieved by 2P-ZOFL.
> > >
> > > 11-	The proof does not follow at all from standard FL approach. There are challenges related to zero-order information and challenges related to proving the exact gradient converges while using zero-order information. Unlike stochastic first-order gradients, zero-order estimators offer a biased estimation of the gradient, meaning that the expectation is not just the exact gradient, but with some additional stochastic perturbation. Moreover, the variance of one-point estimators with respect to the exact gradient is unbounded. These issues make the analysis of convergence and finding conditions on the parameters like the step sizes a bit complicated. Recall that the nonconvexity of the loss function makes the analysis also more challenging. We overcome it to prove the theoretical convergence and find bounds on the convergence rates. The proof is rather technical. We have first to ensure that the gradient bias is vanishing. Then, there is an accumulation of sums due to this bias and one has to ensure the boundness of these terms even when the variance of the gradient estimate is NOT assumed to be bounded (this is different from other two-point zero-order methods, even though not used in FL, that assume boundedness of the variance from the beginning, but due to the use of single point estimate that incorporates the channel perturbation, this assumption cannot be used in our work). We encountered this difficulty by employing doob-martingale inequality (among other technical steps) and without using any convex inequality, which is actually another main difficulty here.

---

### Official Review · Reviewer_eq2v · 2023-11-03

**Soundness:** 3 good
**Presentation:** 2 fair
**Contribution:** 3 good
**Rating:** 5
**Confidence:** 3

**Summary:**

This study presents a framework that introduces a zero-order method with one-point and two-point gradient estimators. Unlike previous methods, it directly integrates the wireless channel into the learning algorithm, addressing non-convex FL objectives and channel complexities. The authors theoretically prove the convergence of this zero-order federated learning (ZOFL) framework. Furthermore, the authors demonstrate the convergence behavior for both one-point and two-point estimations compared to FedAvg through experiments with different binary distribution scenarios.

**Strengths:**

The authors propose two zero-order gradient estimators for FL, which include the noisy wireless channel in the gradient estimation.

The authors provide a theoretical analysis of their proposed estimators and prove the convergence of their FL algorithms in the non-convex setting.

The authors conduct a comparison of their proposed zero-order FL algorithm with FedAvg.

**Weaknesses:**

In the abstract, the authors provide theoretical convergence results as a function of K as well as throught the main paper without defining what K represents.

In the introduction, the authors use "they," but it is unclear to what it refers. Does it refer to all previous works or to the authors in the work they cite?

In the motivation section, specifically in the "communication bottleneck" paragraph, the authors discuss allowing partial participation to reduce the communication bottleneck. The authors could discuss works that optimized the partial participation to reduce the required number of communication rounds, such as power-of-choice, filfl, and divfl. Even in the experiments, they only compare to FedAvg, while several other variants show fewer communication rounds.

While the authors focus on the non-convex setting in their theoretical analysis, it would be interesting to see the convergence rate in the (strongly) convex setting as well.

I think section 2.3 should precede section 2.2 for better clarity. One needs to understand the estimators of the gradients before delving into the final algorithm.

It is very unclear why the authors only consider binary classification tasks, which seem very simple to learn. This makes it difficult to judge the quality of their proposed solution. We need to see if their proposed algorithm works well in settings with multiple classes.

**Questions:**

Can the authors explain why it is particularly interesting to train clients in the wireless setting and include the noisy channel in their estimation rather than using wireless communication protocols to encode and decode messages (if needed) and then conducting FedAvg or variants of FedAvg? By construction of the algorithm, the clients send much less but much more frequently. Why and when is this a more interesting approach?

Why the algorithm only considers even integers k?

In section 2.3, Eq. 3 and Eq. 4 do not include the terms "d/2*gamma" and "d/gamma," respectively, as defined in the introduction. Can the authors explain the reason for that?

While their proposed solution is a zero-order method, why the authors did not compare to previous zero-order methods as well as other variants of FedAvg with optimized client participation.

Can the authors explain why they only consider binary classification tasks in their empirical results? Will their approach perform well in scenarios with multiple classes?

---

> ### Author Response · Authors · 2023-11-22
> **Comment (1/2)**
>
> We thank the reviewer for taking the time to read the paper and writing a review.
>
> In reply to the weaknesses:
>
> 1-	Thank you for bringing this to our attention. $K$ is the total number of iterations until convergence.
>
> 2-	“They” refers to the authors of the previously referenced work.
>
> 3-	We thought it would be more powerful to assess convergence performance with the ideal case. But we do provide a quantitative comparison in communication efficiency with FedZO in Appendix E.3 whose communication efficiency strategies include partial device participation and many local update steps before uploading.
>
> 4-	We thank you for this remark. Nonconvex setting is usually more practical (especially that most of loss function in ML are not convex). We therefore concentrated this work on the hardest task, which is the nonconvex setting. The convergence analysis is usually more difficult since one cannot use the strong convex inequality in the proof. Given the limited time and page limit for a conference paper, we prefer to keep the focus on the nonconvex case. Otherwise, the analysis in the paper will be very lengthy, and will not meet the page limit of a conference.
>
> 5-	We thank you for this advice. We thought it might clarify the steps of how the gradient estimate is constructed using the algorithm before we introduce it. We were worried that its long structure might feel confusing to the reader. Other reviewers preferred to see the algorithm even before section 2.2.
>
> 6-	The quality of our work is heavily concentrated in its mathematical aspect and not numerical experiments. The simulations served only as a practical example to show the interested reader that the algorithms indeed work. We believe that a mathematical proof of convergence is stronger than simulation, since one can make few tests but cannot simulate all potential settings.
>
> In answer to the questions:
>
> 1-	The communication bottleneck is a real issue of federated learning that can hinder its practical implementation. The delay caused by the queued uploaded information can impede the whole learning process and cause data units to be dropped in addition to causing congestion to other communication applications of the clients using the bandwidth. An important note here too is that decoding information requires estimating the channel scaling at every iteration and that can be time and resource-consuming and add to the already existing congestion. This is the interest of the alternative that we provide as it solves both these issues, scalar values consume nothing in comparison to the needed resources; The exchange is not “much” more frequent, and even if it was, the transmitted information size still scales as $O(1)$ and not as a $O(d)$ and this makes all the difference.
> This approach is more appealing when the dimensions of the gradient become much greater than the channel capacity. Our proposed algorithm provides a useful and efficient alternative for applications requiring online and seamless implementation. And like any other solution in research, the conditions of the problem at hand dictate the preferred method and the type of compromises to make.
> As a side note, coding and decoding messages in fact add noise to the transmitted and received information via quantization and channel estimation errors, so there isn’t really any perfect transmission, and that is rarely considered in the design of standard algorithms.
>
> 2-	The algorithm runs on even integers k since there are two reception steps for every k, the first reception is when the channel introduces a scaling a $h_{i,k}$ to user $i$’s signal, and the second reception is when the scaling is $h_{i,k+1}$. We may run $k$ on all integers and consider the second scaling as $h_{i,k+\frac12}$, but we prefer the notation we used in the paper.
>
> 3-	The standard gradient estimates include the factors $\frac{d}{2\gamma}$ and $\frac{d}{\gamma}$, respectively, as their expected value is analyzed using Stoke’s theorem to say that the estimator is an unbiased estimate of the gradient of a smoothed version of the objective function and thus a biased estimate of the exact gradient of the objective function. In our analysis, we use Taylor’s theorem to do the analysis and show the bias and so, this is why the form differs a bit. Both forms are acceptable as long as the step sizes are properly chosen.

---

> > ### Author Response · Authors · 2023-11-22
> > **Comment (2/2)**
> >
> > 4-	In convergence performance, generally zero-order methods cannot have a better rate than the rate we proved in our paper under the same assumptions on the objective function (smoothness, nonconvexity). This is why it makes sense to compare it when first-order information is available. In (McMahan et al., 2017), the best performance of FedAvg is for E=1 local updates, this is why we chose to compare the convergence performance with the “ideal” case. If we were to compare with greater values of E in terms of communication exchanges (i.e., with optimized client participation), we can simply divide the number of symbols needed by the number of clients chosen at each round, similarly to what we did in Appendix E.3.
> >
> > 5-	As already mentioned in the previous section point (6), the simulations were only meant as a numerical example.

---

### Official Review · Reviewer_ufXa · 2023-11-06

**Soundness:** 2 fair
**Presentation:** 2 fair
**Contribution:** 2 fair
**Rating:** 3
**Confidence:** 4

**Summary:**

The paper uses zeroth-order optimization in federated learning over wireless channels with unknown channel gains. The main advantage is to reduce the communication overhead, since clients only need to transmit scalar values to the server in the case of zeroth-order optimization.

**Strengths:**

- The reduction of client-to-server communication by leveraging zeroth-order optimization is interesting.

**Weaknesses:**

- The main Algorithms 1 and 2 rely on Equations (3) and (4) to provide gradient estimates. However, Equations (3) and (4) do not provide estimated gradients due to the unknown channel gain $h\_{j,k}$ and noise $n_{j,k}$ terms. The channel gain and noise can drift the parameter update $\Phi_k$ to arbitrary directions. Since they are unknown, the direction of the gradient remains unknown and the multiplication of $\Phi_k$ in Equations (3) and (4) may not yield a gradient vector in the correct direction. With possibly incorrect gradient estimates, it is unclear how the algorithms can converge to the correct solution.
- The system model multiplies the channel gains directly with the values transmitted by clients. It seems some kind of analog transmission without channel coding is considered. However, in practice, all cellular communication nowadays use digital communication with encoding, where the resulting bit error or noise will have very different mathematical expressions. The current system model and result does not seem to extend to such practical systems. It is further unclear why there is no channel considered in server-to-client communication (Line 3 in Algorithms 1 and 2).
- The experiments are simplified and do not really have a baseline that is compared with, since FedAvg runs in an idealized setting without channel effects. The paper should compare with baselines in the same system setup (i.e., with channel effects of the same statistics). Some well-known FL algorithms with communication efficiency, such as top-k parameter compression with error feedback, should be compared with too.
- Only very simple binary classification tasks using MNIST and FashionMNIST datasets and simple models have been considered in the experiments. It is not clear how the algorithms perform with more advanced datasets and models.
- The writing of the paper needs significant improvement. To me, the main contributions (and especially the usefulness of such contributions) remained unclear before page 5, while pages 6-8 include mostly a list of mathematical assumptions and results with only a limited amount of explanation on what it is useful and novel.

**Questions:**

Please try to address the concerns mentioned in Weaknesses above.

---

> ### Author Response · Authors · 2023-11-22
> **Comment (1/2)**
>
> We thank the reviewer for taking the time to read the paper and writing a review. To address your concerns:
>
> 1- While we appreciate your effort in reviewing our work, we respectfully don’t agree with your claim especially since we provided solid mathematical proof that our estimator is a biased estimator of the gradient and that our algorithm converges. The fact that there are perturbations (channel impact, …) and noises that affect the loss function, does not mean at all that one cannot obtain a biased estimation (biased and not unbiased estimation). It is important to note here that these perturbations affect directly the zero-order information (loss function) and we used the observation/concept that by perturbing the loss function in a smart way, one can obtain a biased estimation of the gradient. This is related to the research field of zero-order optimization and this is the main intuition of why the algorithm works (Flaxman et al., 2004). Of course, the presence of channels and other noises (in addition to the nonconvex setting) makes our analysis challenging. We invite the reviewer to examine the appendix for elaboration, as we prove that equations (3) and (4) are indeed biased gradient estimators using the properties of all stochastic elements involved (and we show that the bias vanishes with time), and we prove mathematically that the algorithms converge. Our paper is not just work based on intuition, it is based on mathematical grounds. We invite the reviewer to check the proofs and we will be happy to answer any comments/questions about them.
>
> 2-	In general, channel coding protects the bits (and thus the transmitted information) from stochasticity. Here, our work “embraces” this stochasticity and seamlessly includes it in the learning. This does not mean that what we’re doing is not practical. Our work can be used in current systems. In fact, nowadays, more analog communication is encouraged, especially with mmWave and THz communications, we want to avoid using analog-digital converters as they are very costly. In our work, no such converters are needed, which makes our system cheaper to implement and we don’t need to protect our data. Our estimators are robust to noise and perturbation. This is why what we propose is actually practical. For information, there is an increasing interest nowadays in performing ML over wireless systems by using analog communications, due in part to the reason mentioned above. We invite the reviewer to check the references [a], [b], [c], [d] provided below (even if the setting is different and they use gradient information while we use zero order method, they focus on analog communication).
>
> In the description of 1P-ZOFL, we explain that there’s a channel considered in the downlink server-to-client communication, we just represent it within the stochastic variable $S$ for easier comprehension of our algorithm.
>
> 3-	If we compare against algorithms with the same channel effect or with compression, the performance of the algorithm we compare against will be much worse than the idealized FedAvg. 2P-ZOFL is shown to finally converge to the same result as the idealized FedAvg and 1P-ZOFL shows a highly comparable performance. In other words, we believe the comparison we provided in the paper is stronger than what is suggested by the reviewer. However, we do provide a quantitative comparison in communication efficiency with FedZO in Appendix E.3 whose communication efficiency strategies include partial device participation and many local update steps before uploading.
>
> 4-	The datasets we used are standard in the research to test new optimization/ learning techniques. The simulations only serve as a numerical example to show that the methods work. We believe that a mathematical proof of convergence is stronger than simulations, since one can make few tests but cannot simulate all potential settings
>
> References:
>
> [a] G. Zhu, J. Xu, K. Huang, and S. Cui, "Over-the-Air Computing for Wireless Data Aggregation in Massive IoT," in IEEE Wireless Communications, vol. 28, no. 4, pp. 57-65, August 2021, doi: 10.1109/MWC.011.2000467.
>
> [b] G. Zhu, Y. Wang and K. Huang, "Broadband Analog Aggregation for Low-Latency Federated Edge Learning," in IEEE Transactions on Wireless Communications, vol. 19, no. 1, pp. 491-506, Jan. 2020, doi: 10.1109/TWC.2019.2946245.
>
> [c] W. Fang, Z. Yu, Y. Jiang, Y. Shi, C. N. Jones and Y. Zhou, "Communication-Efficient Stochastic Zeroth-Order Optimization for Federated Learning," in IEEE Transactions on Signal Processing, vol. 70, pp. 5058-5073, 2022, doi: 10.1109/TSP.2022.3214122.
>
> [d] K. Yang, T. Jiang, Y. Shi and Z. Ding, "Federated Learning via Over-the-Air Computation," in IEEE Transactions on Wireless Communications, vol. 19, no. 3, pp. 2022-2035, March 2020, doi: 10.1109/TWC.2019.2961673.

---

> > ### Author Response · Authors · 2023-11-22
> > **Comment (2/2)**
> >
> > 5-	We feel sorry that the reviewer found the contribution unclear in the introduction. We cannot, however, present the algorithm and the proposed estimators before the introduction. We’ve tried to the best of our abilities to describe the setting of the paper (describing what zero-order estimators are and the scaling wireless networks cause), the issues in this setting, and what solution we propose vs the already proposed solutions. In the subsection “Challenges and contribution”, we present challenges very specific to the subfields involved and what we offer to resolve them. We don’t understand what would make the contribution clearer.   Moreover, the math we provide on pages 6-8 is central to our proofs of the convergence that we invited the reviewer to check carefully in point (1).  Aside from proposing a new method, our main contribution is proving that it converges mathematically. All assumptions are necessary and useful, and we explicitly state that the use of the stochastic noise is what’s novel to prove the almost sure convergence of the exact gradient and not just its expectation in nonconvex zero-order optimization.  The main novelty, however, is in the algorithm and estimators themselves which are based on scalar exchanges and include the channel in the learning. We will provide more explanations in the final version.

---

### Official Review · Reviewer_3zxv · 2023-11-07

**Soundness:** 2 fair
**Presentation:** 2 fair
**Contribution:** 2 fair
**Rating:** 3
**Confidence:** 4

**Summary:**

This paper proposes algorithms for one-point and two-point zero-order gradient estimators for federated learning, which relies on querying function values. This is in contrast to first-order and second-order methods for federated learning. The paper claims to be the first method that involves the effects of the wireless channel without explicitly requiring the knowledge of the channel state coefficients. Finally, the paper provides theoretical and experimental evidence for convergence and provides an upper bound on the convergence rate for both their one-point and two-point estimators.

**Strengths:**

To my knowledge, the application of zero-order one-point and two-point estimates without explicitly estimating the channel state coefficients is novel in federated learning.

The algorithm and theoretical contributions are not trivial and warrant more experimentation to determine the relative performance among other competing methods which also claims to save communication resources.

**Weaknesses:**

The experiment set-up is questionable in comparison to existing work, especially when compared to the experiments done in the baseline FedAvg federated learning algorithm. It is not clear why the authors decided to pick only two digits “0” and “1” from the MNIST dataset and only “shirts” and “sneakers” from the FashionMNIST dataset. On the other hand, the experiments done in the original FedAvg paper were done on the full multi-class datasets, such as MNIST and CIFAR-10.

The results were also only compared to FedAvg, with no comparison done against the competing methods (which also use less communication resources) cited in section 1.1.

The inconsistency in the 2 examples in Section 4 (Experimental Results) makes the discussion unconvincing without a supporting explanation or ablation study. For the first example, a logistic regression model is used, and the images were preprocessed using a lossy autoencoder, and was based on 2 class-labels from MNIST. The second example uses a different model, without compression and on a different dataset. The results would have been more convincing if the set-ups on the 2 dataset were similar. As it stands, I am unsure if the differences between the 2 examples, as illustrated by Figure 2 and 3, are due to the different dataset or the difference in compression or the different type of model used.

The experiment results and short discussion left much to be desired. It is not clear what the conclusions are from Figure 2 and Figure 3. Are the number of communications rounds the bottleneck or the number of scalar values the bottleneck? It would be better to provide a clearer discussion of the main advantage of the methods proposed in the paper.

**Questions:**

1. Why were the experiment set-ups reduced form (only binary labels) of the MNIST and FashionMNIST? Are the results for ZOFL methods limited to experiments for binary classification tasks?

1. Could the evaluation for the 2 examples (Figure 2 and Figure 3) use a similar model and preprocessing? This would help isolate the reason for the difference in performances. For instance, 2P-ZOFL in Figure 2 shows a much faster rate of convergence initially, when compared to FedAvg, but this is not the case for Figure 3. It is not clear if this means that 2P-ZOFL only converges faster in terms of communication rounds only when logistic regression is used and not when multilayer-perceptron is used.

1. For clarity, can the authors provide the main metric of consideration? Are the number of scalar values more critical than the number of communication rounds in the context of the federated learning example? Perhaps it would be clearer if results on the time taken and capacity of the wireless link is provided.

1. In Section 1.1, under communication bottleneck, several competing methods that save communication resources were cited. How do these methods compare to 1P-ZOFL and 2P-ZOFL?

1. It seems that the proofs hold for a general class of perturbation vectors as stated in Assumption 2, beyond vectors that only consists of 2 unique values for every dimension of the vector (in Appendix E.2), which is interesting. Are there experimental results for these vectors?

1. The results were also only compared to FedAvg, with no comparison done against the competing methods (which also use less communication resources) cited in section 1.1.

---

> ### Author Response · Authors · 2023-11-22
>
> We thank the reviewer for taking the time to read the paper and writing a review. We're grateful for the thoughtful insights provided.
> We must first note that the simulations we provide serve only as numerical examples to show that our method works. They’re not the main contribution. Our main contribution is proposing a new estimator and a new method and providing mathematical grounds for this method. This is the difficult part, and we were able to prove it. We believe that a mathematical proof of convergence is stronger than simulation, since one can make few tests but cannot simulate all potential settings.
>
> To address your concerns in the weaknesses section:
>
> 1-	The used classes for the binary classification were chosen at random. The algorithms work equally well for all other choices of classes. We’re aware that FedAvg was originally used to classify multi-class datasets. However, there’s an important difference to understand when comparing zero-order methods to first-order ones and the amount of stochasticity zero-order adds to the problem. While it is possible to classify more than 2 classes and the number of classes can grow up to a certain number, the more classes there are, the more the stochasticity “blurs out” the precise differences in the features that characterize the classes, especially with one-point estimators that have unlimited variance. This is why it’s rare to find papers applying zero-order for multi-class classification. And the fact that we were able to classify images using zero-order is already important and challenging (even with only binary choices), as the features characterizing images are generally a lot more intricate than other types of data and can easily be distorted by the stochasticity. Classifying other types of data is much easier and more fluid. This is a problem with zero-order in general and not specific to our method. We can however provide other examples of classification and non-classification problems if necessary.
>
> 2-	In the simulations, we only mean to test the convergence performance of our algorithm. This is why we chose to compare with FedAvg, or the “ideal” case. While other methods provide other strategies to save communication resources, their convergence speed/performance is generally worse than FedAvg. However, we do provide a quantitative comparison in communication efficiency with FedZO (alongside FedAvg) in Appendix E.3 whose communication efficiency strategies include partial device participation and many local update steps before uploading.
>
> 3-	We thank you for bringing this point to our attention. We note again that this choice of datasets is random and we only meant to provide different objective function examples. The difference between the two results is indeed due to the type of method used to classify and not the dataset.  In our test runs, classifying images from the MNIST dataset using multilayer perceptron gave the same performance as FashionMNIST. And as can be seen from the figure, FedAvg is also affected by the method used.
>
> 4-	The bottleneck is due to the number of scalar values (symbol) per communication round and not the number of communication rounds, as those scalar values can accumulate from one round to another, causing a major delay and impediment on the learning process itself and that’s the main issue in federated learning.
>
> To answer your questions:
>
> 1-	Indeed the results are not limited to experiments of binary classification and numerous other examples can be used.
>
> 2-	 Certainly! However, with the limited time provided for the rebuttal, we weren't able to run all simulations and provide the results.
>
> 3-	As explained above in point (4), the number of scalar values per communication round is what’s critical in the context of federated learning.
>
> 4-	Please refer to point (2) above.
>
> 5-	The probability distribution of the direction chosen for the gradient estimate can be any symmetrical distribution. For example, Flaxman et al. (2004) use a uniformly random unit vector and Duchi et al. (2015) use a Gaussian random variable.
>
> 6- Please refer to point (2) above.

---

### Official Review · Reviewer_hpcJ · 2023-11-10

**Soundness:** 2 fair
**Presentation:** 2 fair
**Contribution:** 2 fair
**Rating:** 3
**Confidence:** 3

**Summary:**

This paper considers Federated Learning over wireless channels and proposes a zero-order FL method with one-point and two-point gradient estimators. Only scalar-valued feedback from the devices to the server is considered and the effect of the wireless channel is incorporated in the learning algorithm. Theoretical results in terms of convergence guarantees are provided with some experimental evidence.

**Strengths:**

The main strengths of the paper are the following:

1) The paper considers some realistic bottlenecks of implementing FL algorithms over wireless channels, e.g., IoT applications and introduces some new ideas in zero order optimization that do not require calculation of gradients at the devices.

2) With the assumptions made in the paper, the convergence analysis for the proposed 1P and 2P ZOFL algorithms seems concrete.

**Weaknesses:**

The paper has the following weaknesses:

1) The wireless channel model assumed in this work is highly simplistic. Eq. (1) refers to a "flat fading" channel which completely ignores the effect of multipath and inter-symbol interference (ISI) caused by it, which requires more sophisticated processing at the receiver to mitigate its effect. One cannot simply assume a simple channel model that is unrealistic in order to admit tractable analysis of FL algorithms. Furthermore, in the case of IoT devices for which FL methods are applicable, the channel conditions can vary a lot from more stationary in time to highly time-varying.

2) In the 1P-ZOFL algorithm, how does the device know the value of $\sigma_h^2$ to be able to send $1/\sigma_h^2$ to the server? This assumes that the device has knowledge of the wireless channel for the link from itself to the server, which in practical scenarios is only possible if the base station has transmitted pilot/reference signals to the device in prior communication rounds. However, the paper does not specify this at all. Also, the terms $h_{i,k}^{DL}$ are not defined when they first appear.

2) Secondly, most IoT devices (e.g., smartphones, sensors, etc.) as well as the server (which is likely co-located at the cellular base station) have multiple antennas (i.e., MIMO technology) which make estimating the wireless channel not equivalent to estimating a single real-valued scalar value. This may render some or all of the derivations regarding the convergence analysis to be inapplicable.

3) The experimental evaluation is insufficient. It needs evaluations beyond binary image classification. Also, evaluation on more realistic channel models, e.g., those specified by 3GPP such as CDL or TDL models would be helpful to observe the degradation in the proposed algorithms when the assumptions on the channel do not hold.

**Questions:**

Some comments regarding writing:

1) Please refrain from using abbreviations such as "it's", "there's", etc.

---

> ### Author Response · Authors · 2023-11-22
>
> We thank the reviewer for taking the time to read the paper and writing a review. We also appreciate the acknowledgment and understanding of our contribution.
>
> To reply to the weaknesses section:
>
> 1.	We used a standard statistical channel model that is widely used in wireless communications research (textbooks and thousands of IEEE papers). We kindly ask the reviewer to refer to all the references we cited (Yang et al., 2020; Amiri & Gündüz, 2020; Sery & Cohen, 2020; Guo et al., 2021; Sery et al., 2021) and thousand other IEEE papers on wireless and cellular communications that use similar statistical channel models in their research. As an example, we also invite the reviewer to refer to the reference [*] provided below.
>
> In addition, we do consider that the channel conditions are time-varying, we’re only assuming the channel to be constant during the transmission of one (scalar) symbol. To have a general channel model, we consider that the channel is time correlated (but not constant). We believe this a realistic assumption and widely used in wireless systems.
>
> In regards to ISI, in our case, there’s no ISI. Every user transmits only one symbol per upload and waits for the server to send the aggregated model, so there’s no “inter-symbol” interference.  After that, all users’ symbols are aggregated by the server, making use of and keeping the stochasticity introduced by their channels. Furthermore, ISI is a widely studied problem in many IEEE papers and several solutions exist (guard interval, ….), and we believe it does not have to be studied in an ML paper.
>
> 2.	$\sigma_h^2$ can be estimated beforehand and not every instant. As a side note, we use the variance $\sigma_h^2$ just to normalize the received signal, but its presence is not necessary and does not affect the convergence of the algorithm nor the convergence rate if we were to remove it. We apologize for the confusion, $h_{i,k}^{DL}$ is the channel perturbation on the downlink server-to-user transmission signal.
>
> 3.	MIMO only changes the dimension of the channel. It’s equivalent to having multiple copies of the signal, so considering MIMO antennas will induce an additional summation over all the antennas. We must also keep in mind that we’re only transmitting one symbol per user. Considering MIMO antennas does not change anything about our work and contribution.
>
> 4.	The simulations we provide are only numerical examples to show that our method works. They’re not the main contribution. Our main contribution is proposing a new estimator and a new method and providing mathematical grounds for this method. This is the difficult part, and we were able to prove it. We can include other examples if necessary.
>
> Finally, we thank the reviewer for the raised remarks. We would also like to stress that focusing too much on wireless questions/details for an ML paper would somehow lead to an unfair treatment of the paper, as compared to other papers in this area. Thousands of papers don’t use these channel models and our central problem is not a wireless problem. We employed standard models and issues related to the modeling itself are beyond the scope of this paper. These issues have already been studied and resolved. We hope our reply offers clarification.
>
> In answer to the question, we thank you for pointing this out. We will fix it.
>
> Reference:
>
> [*]  Emil Björnson and Luca Sanguinetti, “Making Cell-Free Massive MIMO Competitive with MMSE Processing and Centralized Implementation,” IEEE Transactions on Wireless Communications, vol. 19, no. 1, pp.77-90, January 2020.
>
> This paper has received the prestigious IEEE Marconi Prize Award in Wireless Communications in 2022. We used a similar channel model, and we invite you to check equation (7), where one can see the used channel model.

---

### Meta-Review · Area_Chair_5KfP · 2023-11-22

**Metareview:**

The authors have examined the federated learning challenge in the context of wireless channels. Their proposed zero-order method optimizes this process by directly incorporating channel characteristics into the algorithm. The provided analysis includes convergence studies and experimental validation of the proposed algorithms.

Strengths: The paper's convergence analysis for the proposed one-point (1P) and two-point (2P) Zero-Order Federated Learning (ZOFL) algorithms appears solid under the given assumptions. The application of zero-order one-point and two-point estimates without explicitly estimating channel state coefficients in federated learning is identified as a novel contribution.

Weaknesses:
1. Simplistic Wireless Channel Model, limited applicability of the setup
2. Limited detailed motivation, e.g., Why the choice of zeroth-order optimization in the federated learning context, especially when stochastic gradients are typically readily available?
3. The potential drift in parameter updates and the unclear extension to practical digital communication systems raise doubts about convergence to correct solutions.
4. The convergence bounds are not related to other results in the literature such as with FedAvg, FedProx etc.
5. The experiments focus on binary classification tasks with basic datasets and models, limiting the generalizability of results.

**Justification For Why Not Higher Score:**

All the reviews exhibit a consistent perspective.

**Justification For Why Not Lower Score:**

N/A

---

### Decision · Program_Chairs · 2024-01-16

Reject